# Setting Up Our Lab-in-a-Box: Paving the Road Towards Remote Data Collection for Scalable Personalized Biometrics

**DOI:** 10.3390/jpm15100463

**Published:** 2025-10-01

**Authors:** Mona Elsayed, Jihye Ryu, Joseph Vero, Elizabeth B. Torres

**Affiliations:** 1Psychology Department, Rutgers University, New Brunswick, NJ 08854, USA; jihyeryu@mednet.ucla.edu (J.R.); jgv22@psych.rutgers.edu (J.V.); 2Center for Cognitive Science, Rutgers University, New Brunswick, NJ 08854, USA; 3Computer Science Center for Biomedicine Imaging and Modeling, Rutgers University, New Brunswick, NJ 08854, USA

**Keywords:** scalable research, mobile technology, digital biometrics, motor biomarkers, electrocardiography (ECG), wearable sensors

## Abstract

**Background:** There is an emerging need for new scalable behavioral assays, i.e., assays that are feasible to administer from the comfort of the person’s home, with ease and at higher frequency than clinical visits or visits to laboratory settings can afford us today. This need poses several challenges which we address in this work along with scalable solutions for behavioral data acquisition and analyses aimed at diversifying various populations under study here and to encourage citizen-driven participatory models of research and clinical practices. **Methods:** Our methods are centered on the biophysical fluctuations unique to the person and on the characterization of behavioral states using standardized biorhythmic time series data (from kinematic, electrocardiographic, voice, and video-based tools) in naturalistic settings, outside a laboratory environment. The methods are illustrated with three representative studies (58 participants, 8–70 years old, 34 males, 24 females). Data is presented across the nervous systems under a proposed functional taxonomy that permits data organization according to nervous systems’ maturation and decline levels. These methods can be applied to various research programs ranging from clinical trials at home, to remote pedagogical settings. They are aimed at creating new standardized biometric scales to screen and diagnose neurological disorders across the human lifespan. **Results:** Using this remote data collection system under our new unifying statistical platform for individualized behavioral analysis, we characterize the digital ranges of biophysical signals of neurotypical participants and report departure from normative ranges in neurodevelopmental and neurodegenerative disorders. Each study provides parameter spaces with self-emerging clusters whereby data points corresponding to a cluster are probability distribution parameters automatically classifying participants into different continuous Gamma probability distribution families. Non-parametric analysis reveals significant differences in distributions’ shape and scale (*p* < 0.01). Data reduction is realizable from full probability distribution families to a single parameter, the Gamma scale, amenable to represent each participant within each subclass, and each cluster of similar participants within each cohort. We report on data integration from stochastic analyses that serve to differentiate participants and propose new ways to highly scale our research, education, and clinical practices. **Conclusions**: This work highlights important methodological and analytical techniques for developing personalized and scalable biometrics across various populations outside a laboratory setting.

## 1. Introduction

Personalized medicine requires the integration of data from multiple layers of the knowledge network [1]. These range from clinical tests to behavioral assays/assessments, to various ‘omics’ levels of analysis to integrate the person’s unique features across all these layers. They identify targets for treatment as parameters that maximally broadcast the changes that the treatments evoke in the nervous systems’ biorhythms and serve to track the fluctuations of the treatment’s effects over time. A key component of this process is behavioral assessment, which provides the clinician or the researcher an evaluation of the person’s baseline behavioral states and offers ways to track the changes in such states over time. Current methods of behavioral analysis, however, rely exclusively on observation. They miss important fluctuations in the motion segments making up all behaviors. Some of these segments are deliberately performed towards a sensory goal under some level of awareness. They have been broadly studied in motor control [2,3,4,5]. Other segments, however, occur spontaneously and transpire largely beneath the awareness of the person performing the action [6] and of the observer attempting to describe the action and its consequences [7]. The interplay between such different levels of motor control has been studied by our group at great length and used to characterize nuanced aspects of the individual’s behaviors across different levels of functionality of the nervous system. These have ranged from autonomic [8] to voluntary control [9] revealing the core role of the autonomic nervous system in the regulation and prediction of voluntary motor control.

While we have used high-grade sensors to characterize these aspects of human behaviors, the advent of the digital revolution and the recent advancements in wearable biosensors and computer vision-based techniques for face and pose estimation [10] open new possibilities to scale our research beyond the confines of the lab, and to bring more organic, ecologically valid approaches to our research and clinical practices. To that end, we provide an example of how to package our laboratory research methods and literally send them to our participants in a box, a procedure that we can recycle from family to family. Figure 1 offers a high-level picture of the cycle that we follow to bring our lab protocols, in highly simplified ways, to families that can then gain the ability to participate more actively in our research and to contribute with truly individualized knowledge to the improvement and adaptation of our technologies. The novel contributions of this approach to Personalized Medicine are various. We enumerate some of the challenges and opportunities below, although we acknowledge that the list is not exhaustive:

First, we overcome the limitations of having only a biased sample of the population, often due to transportation issues, or economic hardships. Instead of merely sampling from the sector of the population that is aware of our research and has the means to come to our labs, here we offer technologies that enable broadly scaling remote assessment, with high frequency from the comfort of one’s home. This approach tackles the challenges that people in rural areas or underserved, low-income areas face when participating in our research and opens new opportunities to engage them in research aimed at diversifying medical practices. In turn, this scaling of our scientific quests to include underserved sectors of the population has the potential to dampen the various biases that we currently face when developing new technologies and treatments involving trackers of digital health and wellness.

Secondly, using multimodal assessment through the integration of various biosensors and video-based pose estimation at a micro-level of behavior offers the possibility of finding truly personalized solutions. Instead of focusing on a single parameter or body location for assessment, we can multiplex and examine, across different biorhythms, the micro-peaks of the time series data that emerge from these measurements. Co-registering data and scaling it according to individualized anatomical features under our micro-movement spikes approach [9] offers new ways to assess behaviors across a multiplicity of sampling resolutions and sensor signal types. This in turn enables identification of the parameters that maximally broadcast changes in response to treatments. Such optimal outcome parameters are helpful in identifying personalized targets for treatment. For example, in autism, we identified the noise-to-signal ratio or Gamma scale parameter of gait patterns as the indicator maximally broadcasting the treatment’s effects in both a drug clinical trial involving insulin growth factor in SHANK3 deletion and in a behavioral trial designed to evoke motor autonomy in minimally-speaking children. Both personalized approaches, previously performed in the lab or at school settings, can now be carried out remotely from the comfort of the home, using our proposed lab-in-a-box concept illustrated in Figure 1. This is a critical step in maximizing the child’s comfort and dampening the confounding effects that stress can bring into behavioral data.

Thirdly, we do not assume any theoretical distributions a priori. Instead, the parameters of such a multi-layer approach are used in an empirical estimation process that discovers the best continuous family of probability distribution functions that fit the empirical data (best in a maximum likelihood estimation (MLE) sense). We expand on this aspect of the research in the Section 2. By not assuming a theoretical mean and variance, we offer the possibility of discovering nuances in the data that are specific to the person’s nervous system and in this way, can offer a true personalized medicine approach. The multilayered data offers a direct measurement of the person’s individual signatures rather than a one-size-fits-all population-mean/variance approach to the person’s biorhythms [11]. A corollary of this new way of analyzing such data is that across participants, self-emerging clusters appear that directly classify the data. This is without the need for human heuristics, hand-thresholding of parameter ranges and/or ML/AI training. ML/AI training also uses theoretical noise from assumed distributions rather than biologically inspired noise, a feature that may result in model autophagy disease (MAD) [12]. In this sense, our proposed methods provide true biologically inspired noise across multiple biorhythms that could help improve training in current ML/AI techniques.

There are other benefits linked to the advancement of digital phenotyping with off-the-shelf tools, performed on the go. Discrete clinical scores and patient’s self-reports can be better integrated with continuous digital data, to produce clinically informed digital biomarkers of disorders of the nervous system [13,14]. By itself, digital data can be very useful to identify self-emergent clusters of the population. Yet, when labeling these digital data clusters with the person’s clinical information, we can learn the composition of such population subgroups and better interpret both the digital ranges and the clinical phenotypes. This new way of enhancing digital information and complementing the clinical criteria with physiological underpinnings that are now interpretable, is bound to create new lines of scientific inquiry, while also opening grounds for clinical training [15].

One problem with digital data analysis today is that most methods tend to use absolute physical values of data with disparate physical units, rather than standardized relative values amenable to create appropriate normative scales. Most scales in the clinical realm are criterion-based, meaning that they rarely sample the neurotypical population first, to characterize neurotypical ranges (along with their rates of change across the human lifespan) and measure departures from normative ranges and normative rates of change [16,17]. A case in point is the Autism Diagnosis Observation Schedule (ADOS) [18] which is a criterion rather than a normative test. Its scale does not cover the normative ranges, so it is hard to interpret or understand how a given level of “severity”—as defined by their criteria—stands relative to the typical range. In other words, what is typical in neurodevelopment of social interactions, emotional responses, and communication, is not at present considered by this test beyond telling us (rather late) that something is not on track. To correct for such lateness and heterogeneity in criteria, we here show ways to use a relative measurement that is both personalized, while allowing for the self-emergence of physiologically based data clusters in the population, with precise standardized ranges that nonetheless have a physical-unit range traceable back to the raw data.

Ultimately, such work helps us evaluate the ecological validity of our current approaches—as participants are acting and behaving within more naturalistic settings—and allows for large-scale, truly inclusive studies via the use of commercially available wearable sensing technologies that most people already carry around. In this work, we combine several sample studies with the goal of building the foundation for remote research, assessing its feasibility and usefulness in providing meaningful and interpretable data generated by citizen-driven participatory research.

## 2. Materials and Methods

All three studies (Table 1) took place at the Sensory Motor Integration Laboratory of Rutgers University and/or remotely at the home of the participant. The study protocol was approved by the Rutgers University Institutional Review Board (IRB) [Study ID: Pro2020000154] on 6 September 2023. This protocol covers all participant types, location of research and sensors.

### 2.1. Participants

Three studies were conducted involving a mixture of participants who came to the lab or performed the research at home. Figure 1 shows our lab-in-a-box kit along with the steps to deploy the remote research and recycle the materials from family to family. Table 1 summarizes the participant details and data collected for each of the studies. The first one, called FUME (facial universal micro-expressions) Assessment, had a total of 10 typically developing (TD) college-age participants. They participated in a class taught remotely during COVID-19 and consented to perform an in-class exercise to learn sentiment analyses using Zoom video and audio. This will be the first study described here.

The second study included twenty control participants and eight participants with Parkinson’s Disease (PD) who came to the lab for digital phenotyping. The control group was divided by median ranking, into old (above 55) and young (below 55) neurotypical controls (NT). Included in this study was one participant with essential tremor (ET) and another diagnosed with Autism Spectrum Disorder (ASD). These participants also performed a remote test where we registered voice details and performed audio analyses. Further details on additional studies and biometrics from digitized tests involving the patients with PD have been published elsewhere [8]. Here, we use this subset to instantiate the methods.

A third study (Dysregulation Screening) has eighteen participants (10 TD and 8 ASD, including two autistics with SYNGAP). Here, we assessed cardiac data via electrocardiographic (ECG) recordings, as well as facial and movement activity during a variety of tasks. All participants provided informed consent for data collection and video recording approved by the same Rutgers University Institutional Review Board (IRB Study ID: Pro2020000154 approved on 6 September 2023). We note here that this is a blanket IRB protocol to work with digital data and unconstrained motions occurring at the lab, and/or remotely at home, schools and clinics. The protocol includes Health Insurance Portability and Accountability Act (HIPAA) compliance to protect the participants’ privacy (video and biometric data) as well as the participant’s ownership of the video and biosensor data.

### 2.2. Materials and Equipment

The MC10 BioStamp-nPoint system (Lexington, MA, USA, 2008) wireless sensors were utilized to capture biophysical data (cardiac and motor). Figure 2 shows the sensor locations on the body along with the corresponding sample waveform data outputs. Electrical activity was captured via four separate electrodes on each wearable sensor at a sampling frequency of 250 Hz. This included EMG measurements from the bicep of the dominant arm and ECG measurements from the sensor placed on the lead II position of the chest. Triaxial acceleration was recorded for all three sensor locations—chest lead II position, bicep of the dominant arm, and the leg (upper thigh) opposite the dominant arm (for example, on the left leg if the participant is right-handed) at a sampling rate of 31.25 Hz. A charging station along with a Samsung phone were utilized to calibrate the sensors and timestamp the experimental tasks.

A webcam (Logitech—C920, Lausanne, Switzerland) was utilized to capture video data (30 Hz), including facial data, which was later analyzed via the OpenPose and OpenFace programs from Carnegie Mellon groups [19] to assess motor activity and facial expression. A mini tripod for the webcam was also provided. An isolated microphone was utilized to capture voice data. A standard laptop with internet and microphone capabilities was provided if needed. An expandable camera stand/tripod was also provided for the Pointing task of the third study.

### 2.3. Procedures

#### 2.3.1. Study 1—Facial Universal Micro Expressions (FUME) Assessment

This study was carried out with the participation of 10 neurotypical graduate students (3 males and 7 females). The mean age for this sample was 28 with a standard deviation of 5 years of age. The study used a webcam with a built-in mic and the Zoom application to collect data remotely during a virtual class session. The instructions for data collection were given in class through video examples and explanations of Paul Ekman facial universal micro-expressions (FUMEs) and how to project the seven universal emotions by engaging the various face areas, relative to resting state. Amongst the materials were a screen of uniform background color that every participant used and online videos of Vanessa Van Edwards explaining the FUMEs in various ways, e.g., https://www.youtube.com/watch?v=dX7zIpHaAXE (accessed on 20 August 2025).

##### Task to Characterize Facial Micro-Motions Underlying Instructed Emotions

To assess emotional states, we employed the FUME, a line of research advanced by Paul Ekman [20]. We use seven FUMEs extracted from video. We recorded video for a brief 5 s for each affective state including a baseline resting state, followed by anger, contempt, disgust, fear, happiness, sadness, and surprise. The participants first watched a video in class explaining each emotion, with emphasis on facial features involved in each expression, and examples to later produce the expression (from their own sense of each emotion, without real-time imitation).

#### 2.3.2. Study 2—PD Evaluation

The second study involved 30 participants (22 males and 8 females), mean age 59.8 with standard deviation of 12.1 years of age. Of the 30 participants, 18 patients had mild–moderate severity of diseases based on the Hoehn and Yahr classification. The mean duration of disease was 3.0 +/− 3.6 years. None of the patients had dementia. The mean mini-mental status examination (MMSE) score was 29.5 +/− 0.8. The motor Universal Parkinson’s Disease Rating Scale (UPDRS) was applied to determine overall Parkinsonian motor impairment. The mean UPDRS motor score was 15.8 +/− 8.4. One participant exhibited essential tremors (ET) and another exhibited Autism Spectrum Disorder (ASD). The remaining 20 participants were neurotypical (NT) controls, three of whom were elderly (above 55 years of age, categorized as ‘old NTs’). During this study, participants performed a digital profiling session where they performed a variety of motor-cognitive tasks such as pointing (reaching to a target), drawing, and walking.

#### 2.3.3. Study 3—Dysregulation Screening

The third study involved 18 participants (9 males and 9 females), mean age 20.2 with standard deviation of 10.2 years of age. Of the 18 participants, 10 were typically developing (TD) and 8 were autistics. Of the 8 ASD participants, 2 were diagnosed with a SYNGAP1 mutation often associated with autistic traits and delays in neurodevelopment [21]. To assess different areas of functionality, autonomy and control, our lab has designed a series of tasks that, while brief (ranging between 5 s to 3 min), permit the digital evaluation of the participant using commercially available means from the comfort of their home. Every task described below was first assessed with high-grade sensors in the laboratory to determine optimal time windows to use with lower-grade sensors that are commercially available. The combination of brief functional assays and signals from off-the-shelf instruments make our methods highly scalable.

We recorded biorhythms registered by sensors in Figure 2 and Figure 3 and Table 1 along different systems (motor & facial—peripheral, and cardiac—autonomic) and used a compact set of brief experimental assays designed and previously tested in our lab to evoke signals relevant to the assessment of the systems involved at different neurophysiological levels. These spanned from functions that are vital to the person’s survival (such as heart rate variability), to functions that are important for activities of daily living and the overall person’s well-being. The latter included assessment of involuntary motions that are invisible to the naked eye [22]. An example is undesirable involuntary motions at rest while the person is trying to remain still upon being instructed to not move. We can isolate the physiologically relevant signals and measure the volitional control of the brain over the body at rest [22]. Then, we can assess the level of involuntary jitter, tics, and other motions bound to interfere with overall motor control. We will refer to this as *the resting task*.

An important task that enables us to also measure cognitive–motor performance is one that involves the voluntary control of hand motions during pointing and reaching behaviors. The hands can broadcast our wishes and/or even our hesitations to communicate something to another person in a controlled manner. They can also provide insight into the connection between mental intent and physical realization of that intent. This allows us to observe how much of the person’s action is driven by volition, in a deliberate manner, vs. how much is spontaneously supported, automatically—without adding cognitive load [6]. In this regard, we measure within pointing movements to a visual target, not only the forward segment intended to touch the target, but more importantly, we measure the quality of the backward segment—occurring spontaneously (without instruction). This segment of the reach, which is hidden to the naked eye of an observer and often occurring largely beneath the awareness of the performer [6,9], brings the hand to rest and connects the sequencing of the motion to the target with the pauses en route to the next goal-directed segment. We refer to this later as the *pointing task*.

We can also measure the amount of automatic control that a person has over their movements—particularly movements that enable the person to be independent and ambulatory, free of the need to be assisted by another person. For example, measuring how an individual walks independently and performs turns that require balancing the body up against gravity (as well as continuing to have a fluid gait pattern after turning) can inform us about the person’s well-being in daily living. Extensive work in our lab has derived multiple personalized biometrics of gait. A subset of that work, which we reproduce here in different settings, is provided [23]. We later refer to this as the *walking task*.

At the end of the day, sleep is important to replenish our energy and rest from a busy day. During sleep, if resting is achieved, physical volition is less evident than during awake states—although we know from studying athletes and dancers that they mentally rehearse their motions in their sleep and visible patterns of movements emerge in the biosensor signal [24]. They clearly differ from the extent and smoothness of complex patterns of bodily motions when the person is awake. Yet, they serve to reveal the lack of resting the next morning. Given the importance of resting, we include measurements of sleep and refer to this task as the *sleep task*. In this study, the tasks were performed in the following sequence: Resting, Pointing, Walking, and Sleep. Below, we describe the specifics of each one of these tasks in more detail.

##### The Resting Task

For the resting task, the participant is instructed to sit in a relaxed position and avoid excess movement for three minutes. The participant sits approximately two feet away from the webcam such that the center of the face is aligned with the central point of the webcam. For this task, the face and shoulders of the participant should be apparent in the video frame throughout the entirety of the three minutes. The Appendix A points to sample videos of resting at the lab and at the home environment.

##### The Pointing Task

For the pointing task, the participant was instructed to sit (approximately six feet away from the webcam) in a sideways position relative to the camera such that the movement of their dominant arm was not occluded. The expandable tripod was adjusted such that the target (top end of the tripod) was at the participant’s eye-level and at a comfortable distance away from their sitting position. This distance will vary based on the subject’s arm length but should allow enough space for the subject to fully extend their arm and index finger to reach the target, without overstretching (to avoid fatigue). The subject was then instructed to point to the target and bring their arm back to rest about 50 times (enough times to allow for adequate data analysis) in a self-paced manner (usually about 1.5 min). For this task, the entire side-profile of the body should be visible in each video frame, as the pose estimation software is most robust when it is detecting as many body landmarks as possible. The Appendix A points to sample videos of pointing at the lab and in the home environment.

##### The Walking Task

For the walking task, the participant is instructed to remove all equipment/obstacles to create an open space for walking in a circular pattern. The participant utilizes tape to mark the regions (about 8 feet away from the webcam) that are outside the limits of the video-frame with the experimenter’s guidance. The participant is then instructed to walk naturally within the specified region for a period of 5 min, using turning motions to pace back and forth, from the initial to the final location. The Appendix A points to sample videos of walking at the lab and in the home environment.

##### Daily vs. Sleep Task

Daily tasks consist of all the activities the participant is performing throughout the day after completing the main experimental tasks discussed above. For the sleep task, participants are instructed to keep the wearable sensors on and begin the sleep task right before they go to bed for the night until they completely wake up for the day (approximately 6–8 h). Participants are required to mark down in their activity log any periods in the night where they got out of bed, or their sleep was interrupted.

##### Lab-in-a-Box Procedure

First, a box of the equipment and instructions is delivered to the participant’s home (see Figure 1). An initial remote meeting via Zoom is conducted to introduce the participants to the materials along with guidelines for setting up the equipment within their home. This initial setup meeting is necessary to ensure that all sensors are charged, easily tolerated by the participant, and ready for data collection. The participant is also instructed to utilize an open space within their home and to remove obstacles that may interfere with video recordings. Three sensors along with the corresponding phone are placed on a dock to charge before the data collection period.

During the data collection session, the participant is provided with a participant key to access their personalized recording session. With the researcher’s guidance, the participant is instructed to attach the sensors to conductive adhesives and place them on the body regions of interest (Figure 3). One sensor is placed on the left side of the chest in the lead II position to capture the electrocardiographic (ECG) signal. Another sensor is placed on the bicep muscle of the dominant arm to capture electromyographic (EMG) activity. The third sensor is placed on the left anterior thigh to coordinate the sensors’ placement according to body position. Each of the three sensors simultaneously collect physiological and kinematic data (acceleration in XYZ). The participant then provides the experimenter with remote access (via the Zoom application or TeamViewer) to their computer screen, which provides control over the webcam recordings. The participant is then instructed on how to perform the three main experimental tasks: Resting, Pointing, and Walking. For each of the tasks specified above (Resting, Pointing, Walking) the participant must ‘Start’ and ‘Stop’ the sensor recording via an app on the provided phone. For each task, the experimenter also records separate videos via the webcam and uploads the videos to HIPAA-compliant cloud-based storage for retrieval and analysis. Throughout the recordings, care should be taken to avoid excess interference with video capture including low lighting conditions or other people entering the video frame.

After completion of the three motor tasks, the main experimentation session is concluded and the participant is asked to keep wearing the wearable sensors until the next morning, noting down any strenuous activity conducted throughout the day (e.g., exercising) until they go to sleep. This portion of the data is labeled as Daily Tasks.

Right before the participant goes to sleep for the night, they are instructed to start the Sleep Task and end this task when they wake up in the morning, noting down any sleep disruptions (e.g., going to the restroom). Completion of the Sleep Task marks the very end of the study session, and the participant is instructed to remove the wearable sensors and dock them to recharge. This allows them to automatically upload the data to cloud-based storage. Once completed, the participant returns all the equipment back to the storage container and ships it back to the research lab.

#### 2.3.4. Data Analysis

Micro-Movements Approach—Statistical framework for each biophysical signal

To analyze the biophysical signals harnessed from the autonomic and peripheral nervous systems, we utilize a unified statistical framework coined Micro-Movement Spikes (MMS) [9,11] (Figure 3). These methods are part of a general Statistical Platform for Individualized Behavioral Analysis (SPIBA) [11], which bridges the gap between clinical/behavioral tests and genomics under the Precision Medicine model [1].

Comparing similarity in trends and statistical patterns across the population while using this standardized approach allows us to characterize and stratify different nervous system (NS) pathologies as they emerge and as they change over the lifespan of human development. Biorhythmic signals give rise to time series data with fluctuations that change in timing (inter-peak intervals/peak width) and amplitude (peak heights and peak prominences). In this approach, we have normalized the peaks relative to their overall empirically estimated mean and scaled out allometric effects due to anatomical disparities across participants. Such differences in anatomical lengths impact biorhythmic data and produce undesirable confounding effects that we isolate using the ‘*micro-movements spikes*’ or MMS as our signal (see below, Equation (1)). Importantly, this data derivation from the raw time series data keeps the timestamps and as such enables full recovery of the original data—an important step which is the subject of a different paper.

The MMS are examined with an eye for the best continuous family of probability distributions that can most likely characterize them, i.e., in a maximum likelihood estimation (MLE) sense. Across biorhythms, we have found that the continuous Gamma family of probability distribution functions has best characterized the MMS with 95% confidence. In this sense, we use the Gamma parameters, the shape, the scale, and the Gamma moments (mean, variance, skewness and kurtosis) to characterize the stochastic variations in the multi-biosensor data that we register from the nervous system.

Various parameter spaces and ratios are used to that end, while considering that (1) the noise-to-signal ratio (NSR) of the Gamma family is the scale parameter. This is so because given the shape *a* and the scale *b*, the Gamma mean is $\Gamma_{\mu }=a\cdot b$ and the Gamma variance is $\Gamma_{\sigma }=a\cdot b^{2}$ whereby the Gamma NSR is given by the variance/mean, NSR=a⋅b2a⋅b=b and (2) there is a scaling power law relating the Gamma shape *a* and the Gamma scale *b* with (human) ontogenetically orderly maturation expressed across voluntary motions [9]. Whenever the intent matches the physical consequence of the action under volitional control, this scaling power law is expressed such that as the Gamma NSR (scale) decreases, the Gamma shape increases (towards the Gaussian regime) [9]. This is what naturally occurs during typical human motor maturation. In contrast, as the Gamma NSR increases, the Gamma Shape decreases towards the memoryless exponential regime of the continuous Gamma family where the shape value is 1. This pattern of random noise is uniquely found in autism and neurodevelopmental disorders that are well characterized very early in neurodevelopment by motor disruption [9,25]. These two findings make the NSR our parameter of interest, since given (1) and (2) above, knowing the Gamma scale (the NSR) allows us to infer the Gamma shape and all other moments with high accuracy. The MMS then follow special standardized features that enable us to summarize these entire multimodal time series within a handful of parameters. Specifically, the personalized approach to human motions embedded in natural behaviors identifies the Gamma NSR (scale) as a target for treatment maximally broadcasting change as a treatment outcome or signaling derailment of the nervous systems’ functions.

##### Motor Activity (Wearable Sensors Accelerometer Data)

The accelerometer data from the Inertial Measurement Units (IMUs) were analyzed using the MMS and Gamma methods. Figure 3A shows the locations of the sensors and the sample’s resulting ranges for each location (pectoralis, dominant arm, and dominant leg) for the case of right-handed participants. IMU triaxial accelerometry is measured in g-units whereby 1 g is 9.8 m/s^2^. Figure 3B shows the time series of the pectoralis, arm and leg locations in a sample participant.

Upon peak detection, the overall empirical distribution mean is determined using MLE methods. Upon empirical estimation of the best continuous probability distribution family, we subtract from the empirically estimated Gamma mean the value of each point in the time series. We obtain the absolute deviations from the Gamma mean at each time point and obtain the time series of such absolute deviations (a positively rectified waveform that preserves the original time stamps).

The peaks in this time series of positive deviations from the Gamma mean are normalized to scale out allometric effects due to anatomical disparities across participants. We use Equation (1) to that end. Equation (1) uses the peaks and valleys of the time series. Then, these standardized, unitless peaks are also subjected to empirical estimation. Since they are deviations from the empirically estimated mean, the 0-values are at Gamma mean speed level. We find once again that the Gamma family with shape and scale parameters estimated with 95% confidence according to MLE, best fit the MMS, i.e., the normalized peaks of this time series. Importantly, using this approach, we make no theoretical assumptions of the underlying random process describing these peaks and valleys and we preserve the original timings of the peaks.

Figure 3C depicts the mean deviations. The peaks of these absolute deviations are highlighted as red dots, and as mentioned above, the time stamps from the original raw data at the peak deviations are conserved. Equation (1) provides the normalization to account for allometric effects [26]:(1)$$MMS=\frac{Peak}{Peak+Avrg_{\min\;\mathrm{to}\;\min}}$$

As explained above, in this equation, the local peak (*Peak*) is the absolute deviation from the empirical Gamma mean amplitude, estimated from the original peaks of the full time series, while the average of all points comprised between the local minima surrounding the local peak (the local maximum) are the *Avrg*_min to min_ term. As the average local acceleration decreases, the overall value of the MMS increases towards 1 (higher stochastic fluctuations from the empirically estimated mean). As the average local acceleration increases, the denominator penalizes the numerator, and the overall value of the MMS decreases away from 1 (lower stochastic fluctuations away from the empirically estimated mean).

##### Cardiac Activity (Wearable ECG Data)

ECG signals include consecutive QRS complexes representing each heartbeat in the cardiac signal. The R-peaks (sharp peaks) within QRS complexes are traditionally used to assess the timing between consecutive heartbeats (known as the R-R or inter-beat interval). Accurately detecting R-peaks is essential for assessing the fluctuations in the inter-beat-interval (IBI) signal and in computing various heart rate variability (HRV) parameters (Figure 4). ECG signals are easily corrupted by various artifacts such as baseline wandering and electrode motion [27]. To pre-process the ECG data, an 8th order Butterworth IIR band-pass filter in the 5–30 Hz range was utilized based on the frequency ranges associated with the QRS complex and improvements in signal-to-noise ratios [28]. A 2nd order Butterworth IIR band-stop filter in the 40–125 Hz range was also used to eliminate excess noise/frequencies outside the range of a typical ECG recording. Butterworth filters were used because they minimize corruption of biorhythmic signals and effectively suppress signal disturbances beyond the passband [29]. After preprocessing, R-peaks are detected via a simple peak detection algorithm in MATLAB (MathWorks 23.2.0.2365128 (R2023b)) software, and the IBI signal is obtained by computing the time between consecutive R-peaks (Figure 4A). Changes in the IBI signal (Figure 4B) are the result of autonomic control via the sympathetic (excitatory) and parasympathetic (inhibitory) NSs, which are informative in assessing states of stress/hyperarousal [30,31]. Such NS activities can be inferred via further analysis of the time and frequency domain metrics of the cardiac signal.

##### ECG—Time Domain Analysis

The MMS analytical approach was used on the IBI series to compute the Gamma scale and shape parameters of the cardiac signal (Figure 4C). Poincaré plots were also used to assess sympathetic and parasympathetic activation via time-domain metrics (Figure 4D). Poincaré plots serve as a geometrical and nonlinear method to assess the dynamics of HRV and are formed via building a scatter of the IBI interval against the preceding IBI interval [32]. The width of the scatter is used to determine the SD1 parameter, which reflects parasympathetic NS activity and is correlated with HF power [31]. The length of the scatter is used to determine the SD2 parameter, which reflects sympathetic NS activity and is correlated with LF power [31].

The PhysioNet Cardiovascular Signal Toolbox was also utilized to assess additional HRV metrics. This open-source toolbox is implemented via MATLAB and is designed to assess ECG signals via standardized algorithms [33]. The raw ECG data (sampled at 250 Hz) was assessed via 30 s windows with 10 s overlap and was preprocessed via the toolbox to obtain a fine-grained set of HRV parameters for the full recording. From this we also obtained a distribution of Gaussian moments (Mean, Median, and Mode RR).

##### ECG—Frequency Domain Analysis

The activities of the parasympathetic NS and sympathetic NS can be inferred from the Power Spectral Density (PSD) and to facilitate further interpretation, placed in correspondence with the time-dependent Poincaré plots of the IBI signal (explained below). The high frequency (HF) component of the PSD (in the 0.15–0.4 Hz range) is associated with parasympathetic activity and a general decrease in heart rate [34]. The low frequency (LF) component of the PSD (in the 0.04–0.15 Hz range) is associated with sympathetic NS activity and blood pressure control [30]. Increases in the ratio between the LF and HF components (LF/HF ratio) have been previously associated with stress and intense exercise [35,36]. The LF/HF ratio reflects the sympatho-vagal balance—the contribution of the sympathetic NS in controlling the heart compared to the parasympathetic NS [37].

The HF, LF, and LF/HF ratio were computed by integrating the PSD over the associated frequency range (Figure 4E). Since consecutive IBIs are not uniformly spaced, the Lomb–Scargle PSD was utilized to assess the frequency domain metrics as it can handle signals that have been unevenly sampled without introducing resampling problems [38]. To better visualize which frequencies (in the HF and LF ranges) dominate the cardiac signal across time, continuous wavelet transforms (CWT) were used to obtain magnitude scalograms of the IBI data (Figure 4F). The wavelet scales were converted from pseudo-frequencies to their equivalent Fourier frequencies to ensure proper interpretation of the frequency content (a conversion that MATLAB’s signal processing toolbox performs internally), and then we used the generalized ‘morse’ wavelet (via the MATLAB signal processing toolbox). The generalized morse wavelets were chosen because they are more versatile in analyzing signals with time-varying amplitude and frequency characteristics [39]. CWTs were used to visualize and evaluate temporal changes in frequency power because they can effectively decompose nonstationary signals and provide a personalized assessment of the cardiac activity. Nonstationary signals are those for which the moments of the distribution shift over time, such as is the case here in ECG signals and other biophysical time series of interest.

##### Facial Activity (Proxy Video Data)

Upon video-capturing the face, either via Zoom or with a webcam, or app (as in Figure 5), we use OpenPose Estimation (for bodily pose landmarks) and OpenFace Software (for facial landmarks) to extract a face grid, *3D* gaze, head orientation and action units across the face muscles. We then analyze the kinematic parameters derived from the motion trajectories of positional data (pixels/time) whereby in 5 s, at 30 Hz, we obtain 150 points for each of the 68 landmark points on the grid across the face. To minimize the impact of movements along the axis of the camera, we have standardized each face using a shape normalization that employs centroid alignment (setting the mean of each face to coordinate (0, 0)) and isotropic scaling, such that the Euclidean distance for each point on each mask has a standard deviation of 1.

We have divided the grid into three regions as per the trigeminal nerve sections, denoted V1 (ophthalmic), V2 (maxillary), V3 (mandibular) (see Figure 5) such that we track the stochastic signatures of variations in each region [40]. Upon examining the emotions (relative to resting state), we can see how people automatically group and what regions their facial expressions occupy on the emotional space spanned by our biometrics. This is done as they project emotions on command, or as they spontaneously react to emotions that they watch, e.g., in a video. The same MMS data type and empirical Gamma estimation process as we applied to bodily kinematics and to the ECG data are applied here. Additional details of the analysis for 126 participants can be found in our recent paper [40]. Here, we focus on a continuous assessment of the facial micromotions rather than on a pre- and post-assessment using a minimum of 150 frames per window.

Briefly, for each of the facial regions, we obtain all the points in the grid corresponding to that region, using the Open Pose estimation map from GitHub (https://github.com/, accessed on 20 August 2025). For each point in the grid and across 150 frames, we obtain the pixel positions over time (5 s) and after smoothing, using spline interpolation, we differentiate the curve and obtain the corresponding velocity fields. For each velocity vector, we obtain the speed scalar quantity per unit time (at uniform time intervals) and this temporal speed profile yields peaks and valleys that we subject to the MMS treatment. The peaks of the facial speed MMS also follow the continuous Gamma distribution family. Across all the grid points of each of the V1, V2, V3 regions, we then pool all the micro-peaks and obtain the Gamma estimates for the shape and scale. Given that their log-log map also follows a scaling power law, knowing the shape allows us to infer the scale (the Gamma NSR). Then, we use the Gamma NSR for V1, V2 and V3 and plot the moments of the Gamma PDF for each of the regions. These points have coordinates given by the estimated Gamma Mean (along the x-axis), the estimated Gamma variance (along the y-axis) and the estimated Gamma skewness (along the z-axis). We set the marker size proportional to kurtosis and for each participant, we represent V1, V2 and V3 as the vertices of a triangle. The area enclosed in this triangle gives a unique sense of the micro-expression corresponding to a given emotion. We then plot all 10 participants on this parameter space spanned by the Gamma moments and can appreciate how they group differently for different emotions.

##### Voice Activity (Proxy Audio Data)

We also collected the mic’s audio data and examined the features of the speech waveform. To that end, the enveloped audio (smoothed curve that outlines the maximum amplitude variations of the fluctuating signal over time) filtered at 4000–7000 Hz (centered at 5600 Hz) was examined by segmenting the data according to distinct phases of the stream. This was done using the MIR toolbox 1.7.2 available in MATLAB (Mathworks, version 23.2.0.2365128 (R2023b)). To that end, we used the *mirfilterbank* function using the ‘***Gammatone***’ decomposition [41]. This filter bank decomposition is known to simulate well the response of the basilar membrane. It is based on the Equivalent Rectangular Bandwidth (ERB) filter bank, meaning that the width of each band is determined by a particular psychoacoustical law. For Gammatone filterbanks, the function *mirfliterbank* calls the *Auditory Toolbox* routines *MakeERBFilters* and *ERBfilterbank*, which is the default choice when calling the function *mirfilterbank*. Figure 6 shows the decomposition of 10 filters (of which #8 had proven useful to distinguish PD subjects from controls [8]).

##### Stochastic Characterization of Upwards (Attack) and Downwards (Decay) Phases of Speech

Upward phases (called attack) were separated from downward phases (called decay). The minima and maxima values of the amplitude of the waveform were extracted, using the overall median amplitude value across the session. All values above the median that corresponded to a local maximum (where the slope from the preceding valley to the peak changed from positive to negative) were identified as maxima. Similarly, all values below the median where the slope from the previous maximum to the valley changed from negative to positive were identified as minima. Figure 7 depicts the waveform and criteria.

Because we wanted to focus on meaningful maxima and minima that were most likely produced by actual speech, rather than pauses and non-speech segments (e.g., breathing, sighing), we used a rather strict criterion to separate the maxima. In cases where multiple maxima were present between two minima, we rendered the maximum value as the highest amongst the multiple local (minor) maxima. Our analytics involved segmenting the attack and decay phases (where the attack phase would run from the local minimum to the subsequent local maximum, and the decay phase ran from the local maximum to the subsequent local minimum). For this reason, it was necessary to have only one maximum between two minima. If multiple local maxima were present between two adjacent minima, we chose the maximum (denoted by the red star in Figure 7A, right panel) as the start and end points of the attack and decay phases.

Upon isolation of the maxima and minima, defining the attack and decay segments, we obtained the slopes of the attack and the decay, to build a time series of such values, in correspondence with those segments. For each segment, the attack slope was computed by taking the slope between the local minimum and the subsequent maximum (denoted by the magenta line in Figure 7B). The decay slope was obtained between the maximum and subsequent minimum (denoted by the cyan line). The absolute slopes were then plotted on a histogram along with their noise-to-signal ratio (NSR). This parameter corresponds to the empirically estimated Gamma scale parameter, obtained with 95% confidence intervals through maximum likelihood estimation (MLE) according to the best-fitting continuous family of probability distribution functions (PDFs), which in this case proved to be the Gamma family. This scale parameter was used to compare the PD and control groups.

We also computed the area under the curve during the attack and decay phases, by taking the Riemann sum during the two phases as shown in magenta and cyan, respectively, in Figure 7C. A stochastic characterization of these areas was plotted in histogram form, and the median values used to compare the departure of signatures from patients with PD relative to controls. Finally, the stochastic signatures of the attack and decay phases were examined in relation to the clinical scores of the memory task. The memory task yielded a “memory score”, which was the average of the longest digits correctly recited during the forward and backward memory tasks.

#### 2.3.5. Statistical Tests

To characterize the statistical features of the distribution parameters extracted from the MMS in biophysical data, we used the non-parametric Kruskal–Wallis test. This is a nonparametric version of classical one-way ANOVA, and an extension of the Wilcoxon rank sum test to more than two groups. We use the Matlab software and paraphrase here the explanation: the Kruskal–Wallis test compares the medians of the groups (more than two groups) of data to determine if the samples come from the same population or, equivalently, from different populations with the same distribution.

The test ranks the data by ordering it from smallest to largest across all groups and taking the numeric index of this ordering. The rank for a tied observation is equal to the average rank of all observations tied with it. The F-statistic used in classical one-way ANOVA is replaced by a chi-square statistic, and the *p*-value measures the significance of the chi-square statistic. The assumption here is that all samples come from populations having the same continuous distribution, apart from possibly different locations due to group effects, and that all observations are mutually independent. We report the *p*-values of the comparison and show box–whisker plots of the data being compared. Bear in mind that we compare parameters of the continuous Gamma family of probability distribution functions best fitting the biophysical data in an MLE sense, along with the corresponding Gamma moments of the standardized micro-fluctuations (the MMS) explained in Section 2.3.4. 

#### 2.3.6. Measuring Distance in Probability Space

In these parameter spaces of interest, points represent families of probability distributions. We use the Wasserstein distance, also known as the Earth mover’s distance, defined between probability distributions on a given metric space. For example, in the project involving face grids whereby we empirically obtain the continuous Gamma family of probability distribution functions to represent the flow of micro-movement spikes’ motions on the ophthalmic (V1), maxillary (V2) and mandibular (V3) regions of the face, we track for each of the seven emotions the pairwise EMD from each emotional trajectory to the others, to automatically isolate the most distinctive emotion emerging in a given situation. This information is obtained for 5 s long windows with 50% overlap and represented in histogram form, in heat colormaps and as continuous curves capturing the evolution of the EMD values.

## 3. Results

### 3.1. Study 1—FUME Assessment (Facial Activity)

The individualized estimation of the Gamma moments for each of the 10 participants and each of the three regions V1–V2–V3 yielded differentiable patterns for each of the seven FUMEs (Figure 8). The markers were joined by a line to form a triangular shape. The larger the area of the triangle, the larger the range of parameters representing the FUMEs. Sadness and Surprise showed three unambiguously separated clusters of participants with no overlapping patterns. Anger, Contempt, Disgust, Happiness and Sadness showed one or two participants with large parameter ranges (large triangular area), while all others had clustering patterns on one area of the Gamma moments space. Figure 8 shows six of the FUMEs.

The representation of the FUMEs in this manner provides a way to ascertain the ranges of facial micro-motions for each of the regions in Figure 5, as each emotion projects on command. For each person, we can then see not only the signatures of emotions on the Gamma moments space but, more importantly, we can visualize and quantify the unfolding of the stochastic trajectories of a given video of spontaneous emotional reactions, as the person’s facial micro-motions display positional excursions over time. These excursions can then be visualized on the Gamma moments parameter space relative to those of the person at rest, or relative to the signatures of another model person (to establish a relative metric, e.g., in dyadic interactions). Figure 9 shows an example whereby the excursion signatures of one person are plotted relative to those of a model person.

Figure 9 then shows the trajectories for V1, V2 and V3 from one of the participants. These regional trajectories are plotted in reference to the FUME signatures of another participant which served as a model to localize them relative to each group of emotions. This example can be extended to a group, as in Figure 8, so we could appreciate the unfolding of one person’s sentiments relative to the signatures of a cohort.

To quantify the differences between the person’s sentiment trajectory and the reference signatures, we use the EMD. This is shown in Figure A1A for each of the face regions under consideration. We provide heat color maps for each of the seven FUMEs along the columns and frames at 5 s windows with 10% overlap along the rows. The color represents the EMD obtained pairwise between the stochastic trajectory frame and the reference signal. Figure A1B provides for each face region, the EMD trajectory (vertical axis) as it evolves along the frames (horizontal axis). Figure A2A shows the frequency histograms of the EMD quantity and automatically separates the emotions in classes according to the reference signature. Figure A2B provides the heat colormap matrix of pairwise statistical comparisons whereby each emotion of the seven FUMEs is compared to all other emotions. Each entry reports the *p*-value with significant differences at *p* < 0.01 level marked by a white star and *p* < 0.05 marked with a white-edged open circle. The color bar shows the *p*-values from 0–1.

### 3.2. Study 2—PD Evaluation (Voice Activity)

The voice analysis from the patients with PD showed significant differences in the area under the curve (*p* < 0.01) and in the Gamma NSR empirically estimated parameters (*p* < 0.05), with differences in the memory scores as well (without reaching statistical significance). These results are depicted in Figure 10 where we compare the signatures of the patients with PD against the age-matched controls, an elderly person with Essential Tremor (ET), an ASD participant, and young controls. The figure depicts the most and least severe patients to help localize the ranges of the parameters of interest: Log(Gamma NSR) along the x-axis, Log(Area under the curve) along the y-axis, and the memory score along the z-axis. Voice analysis revealed marked differences between elderly and young controls, besides differences with the patients with PD. Furthermore, the ASD participant was far from the NTs in terms of age, and closer to the location of the PD and the elderly.

### 3.3. Study 3—Dysregulation Screening (Cardiac, Motor, and Facial Activity)

ECG data was explored in the frequency and time domain as well as via the MMS approach where the Gamma parameters were obtained. We characterized the cardiac activity of each participant based on the Gamma parameters of the extracted IBI series as well as their corresponding magnitude scalograms (Figure 11). We compared each participant across the three main tasks and found that for the resting and walking task, the elderly participant (Subject T03) had a significantly different cardiac profile compared to the two adult participants (Subject T01 and T02) based on the decreased scale (noise-to-signal ratio) and increased shape of Gamma parameters. This demonstrates a higher predictability in the IBI signal. A similar pattern is observed in the magnitude scalograms where subject T03 shows consistent sympatho-vagal balance whereas the others show relatively higher magnitude in the lower frequency ranges. This personalized mathematical characterization can thus be used to help quantify/infer age range differences/developmental change.

A common pattern is also observed when comparing the Gamma and HRV parameters (Figure 11). A high scale (NSR) and low shape parameter is often associated with elevated power in the magnitude scalogram. The lower shape value (which is associated with higher noise) is indicative of ranges of the Gamma PDF closer to the memoryless exponential distribution outputting a random noise IBI signal. This is here related to impaired sympatho-vagal balance whereas a low scale/NSR and high shape is associated with a more stable HRV response. Given the systematicity of this result across other participants [42], the Gamma characterization alone can be indicative of the underlying autonomic stability/activity.

When exploring HRV responses across subjects, we find task-specific clustering based on the level of physical exertion required (Figure 12). For the walking task which requires full body movement, we observe lower Gaussian moments for the IBI series (higher heart rate) along with relatively lower SD1 and HF power (parasympathetic activity). This is followed by the pointing task which requires repetitive arm movements. The resting task, which requires the least amount of movement, shows the lowest heart rate and spans a larger range of values within the frequency and time domain parameters, indicative of better sympatho-vagal balance across subjects.

A similar pattern is observed when comparing the heart rate statistics during daily activities vs. sleep (Figure 13). During the sleep task, we observe relatively higher RR intervals for each subject, indicating a slower heart rate compared to daily tasks, where more movement may take place during the day. This task-specific clustering may be helpful in assessing levels of sleep quality and/or gauging overall levels of physical activity during the day without the need of surveying or obtaining specific details from the participant.

#### 3.3.1. Cardiac Activity—Comparison with Atypical Populations

To assess the translational clinical value of the above methods, we utilize the same analytical pipeline for neurotypical controls, to characterize the ECG signal of participants with atypical neurodevelopment. When exploring the Gamma and HRV statistics of cardiac activity, we can separate typically developing (TD) participants from participants with a diagnosis of autism spectrum disorders (ASD) (Figure 13). Specifically, we find that ASD participants demonstrate cardiac signatures with relatively higher Gamma NSR (explained by higher Gamma variance and lower Gamma mean). This corresponds to a relatively faster heart rate and less predictable cardiac pattern, suggesting sympathetic overdrive at baseline in ASD. Indeed, this is what we find when exploring the Poincaré metrics in the frequency domain—ASD subjects demonstrate relatively higher SD2/SD1 ratios, indicating hyperarousal of the sympathetic system (see Figure 14). When expanding our analyses to include participants diagnosed with SYNGAP and chronic pain (CP), we find interesting patterns that warrant future research see Figure 15 in the next section).

Exploring where these participants fall on the Gamma parameter plane reveals that individuals with ASD as well as those with SYNGAP and CP tend to exhibit higher scale (higher Gamma NSR) and lower shape values (tend to fall in the upper left region of the Gamma parameter plane) compared to TD individuals, who tend to fall on the bottom right region of the Gamma plane. This may indicate some common etiologies between subjects with ASD, SYNGAP, and chronic pain in terms of cardiac activity. Upon further analysis of participants who showed the opposite trend, we find that TD individuals falling on the upper left region of the Gamma plane demonstrate relatively atypical magnitude scalograms while ASD individuals in the bottom right region of the Gamma plane exhibit typical magnitude scalograms. This may indicate that some TD participants may have an underlying sympatho-vagal imbalance while some ASD participants may exhibit more stable cardiac responses. While additional subjects are needed to confirm the observed patterns, such characterization of the cardiac signatures proves useful in stratifying atypical vs. typical states of the HRV. They signal levels of dysregulation and provide a quick visual summary of the heart’s activity.

#### 3.3.2. Motor Activity—Arm Movements During a Simple Pointing Task

When evaluating the acceleration activity across all three body regions (pectoral area of chest, upper arm, and left thigh) we find unique patterns across typically developing (TD) participants. These patterns can differentiate activities and body parts, as indicated by the results shown in Figure 16 for three representative typical subjects.

Extending the question to ASD and TD groups for the Pointing task (Figure 17), we show differentiation of some Gamma parameters, despite small sample sizes. For the upper arm acceleration MMS, we see a significant increase in the Gamma variance for the ASD group (*p* < 0.05) accounting for higher Gamma scale (NSR), which is paired with lower Gamma shape. Based on median and range values, and compared to the TD group, however, these differences were not significant (most likely due to the small sample size).

Interestingly, when evaluating the motor signal (acceleration MMS) of the upper body/pectoral area during the Pointing task, we observe significant differences in the Gamma skewness (*p* < 0.05) and kurtosis (*p* < 0.001) between the ASD and TD group. Such findings suggest that the acceleration movement of the upper arm for the Pointing task is generally noisier and more unpredictable in ASD. This result is in line with previous findings that demonstrate an increased Gamma NSR in the linear speed profile of autistics derived from positional data [9]. The extension to IMUs makes the methods amenable to use in tandem with consumer-grade accelerometers embedded in cell phones [11]—a result that we had previously found useful to separate patients with PD and controls. A larger sample size in both ASD and TD groups and a closer look at the intentional (goal-directed) and unintentional (goal-less) aspect of the arm movements could allow us to further elucidate the motor signatures across groups.

#### 3.3.3. Facial Activity—Resting State Facial Affect and Muscle Movement in ASD vs. TD Participants

When analyzing the facial landmarks obtained across ASD and TD participants visually, we observe excess noise in the facial landmark positions across all three trigeminal regions (V1—ophthalmic, V2—maxillary, V3—mandibular) of the face for most ASD participants compared to TD participants (compare Figure 18A representing a TD participant vs. Figure 18B representing an ASD participant). Such changes are not easily visible to the naked eye from assessing the raw video data. When applying the Gamma approach to each region, we appreciate the differentiation in TD and the collapsing of the signatures in ASD.

For ASD participants, we see that the Gamma scale and shape parameters for all three regions fall nearer to each other on the Gamma plane compared to TD subjects where the V1, V2, and V3 each fall in their own unique location. The Gamma moments and PDFs were often more differentiable across the three regions for TD participants. However, group differences in the Gamma parameters did not follow a distinct trend. This suggests that each participant demonstrated their own unique patterns in facial motor activity, with differences across the V1, V2, and V3 regions being more distinct for TD subjects. Indeed, when evaluating the transfer entropy (TE) between the three facial areas for both the ASD and TD groups, we find a wider range of TE between V1→V2 that is significantly different from the TE between V1→V3 (*p* < 0.05) and V2→V3 (*p* < 0.01) in the TD group (Figure 18E). This difference in TE was not found in the ASD group, with the flow of information being about the same across all three regions of the face (Figure 18F. This result further suggests that there is a lack of differentiation in the motor activity across the V1, V2, and V3 regions of the face in ASD subjects.

## 4. Discussion

In this work, we discuss a new approach that we coined “*Our Lab in a Box*”. We developed this approach to acquire behavioral data remotely, from participants who quite often could not make it to our labs or were uncomfortable performing a study in a new environment. In the context of autism, this new concept of participatory research engages families that, for a variety of reasons, have not been able to come to our university and participate in our lab research on campus. In this sense, we bring the research lab to their homes and from this more comfortable and natural setting, we can acquire the data and have the families collaboratively send it to our lab via secure, HIPAA-compliant channels and under IRB-approved protocols.

We not only designed ways to acquire the data using commercially available digital tools, but most importantly, we designed brief and easy-to-do protocols to evoke the type of activities that we needed to evaluate to assess the functionality of the person’s nervous system, while trying to avoid taxing the person excessively. Many current observational assays take at least an hour to complete, a real challenge for neurotypicals and much more so for people with nervous systems that developed differently or that suffer from neurodegeneration. Using our motor systems functional taxonomy [6], we assess biophysical signals from the autonomic, the involuntary, the spontaneous, and the voluntary levels of functional control. Through our brief protocols, tailored to the person’s needs, we were able to both acquire the data and analyze it using new methods that standardize the outcomes. By scaling out allometric effects due to anatomical disparities across participants, we were able to sample the person from head to toe and derive standardized biometrics on appropriate parameter spaces that automatically separate signatures of dysregulated states from those of regulated states. These included not only those linked to the face and cardiac activity, but also those related to the body in motion, under a variety of behavioral states that ranged from sitting at rest, to automated walking, voluntary reaches, and sleep.

We demonstrate in this work the feasibility of deploying this method remotely using means that are commonly available to most people—whether the person lives in rural areas, in underserved communities, or even abroad, it is possible to take such measurements remotely and establish a highly scalable model of participatory research. Although our small sample sizes across the three preliminary studies serve as a limitation, several interesting results—discussed below—emerged from the small pilot sessions implementing the lab-in-a-box methodology.

### 4.1. Cardiac Activity

Using the standardized MMS data type derived from IBI time series and pairing it with the characterization of the time series as a Gamma process, we were able to separate TD vs. ASD participants. Specifically, we found that ASD subjects demonstrated cardiac signatures with relatively higher LF/HF ratios, driven by lower HF power. Abnormally elevated LF power has been associated with flight-or-fight mode states [43], explained by atypically high sympathetic drive. In tight correspondence with the abnormally elevated LF power, we also observed elevated Gamma scale (noise-to-signal) ratios. Furthermore, because of the tight inverse linear relationship between the log Gamma scale and the log Gamma shape, knowing one parameter, one can infer the other with high confidence. As the Gamma scale increases, the Gamma shape decreases towards the memoryless exponential distribution ranges of the continuous Gamma family of probability distribution functions. This result indicates that at high sympathetic drive, in flight-or-fight mode, the time series featuring the IBIs have elevated random noise, which would greatly depart from the autonomic system’s highly predictive temporal code of a true pacemaker-like signal. The heart being such an important drive for the rest of the nervous system, it is not surprising then that the dysregulated code was reflected as well in other biophysical signals of the face and body.

Exploring further the HRV signals through the Poincaré metrics across groups, we find that the ASD subjects demonstrate relatively higher SD2/SD1 ratios, driven by a decrease in SD1, indicating parasympathetic inhibition and/or hyperarousal of the sympathetic system. It is possible that the so-called “restricted and repetitive behaviors” observed in ASD are mere self-discovered solutions by the autistic nervous systems, to sooth the system through noise cancellation. This in turn would serve as a coping mechanism to increase the HRV signal’s predictability and help relieve the anxiety and distress elicited by the uncertainty that such sympathetic hyperarousal present at baseline creates [9,44].

When expanding our analyses to include participants diagnosed with SYNGAP and chronic pain (CP), we find interesting patterns that warrant future research. Assessing where these subjects fall on the Gamma plane when the activity comes from resting states, we find that individuals with ASD as well as those with SYNGAP and CP tend to exhibit higher Gamma scale values (i.e., higher dispersion of the Gamma PDF, equivalent to the noise-to-signal ratio derived from the MMS peaks of the IBI time series). These patterns are in correspondence with lower shape values of the Gamma PDFs, tending to the memoryless exponential regime of the Gamma family. This is in marked contrast to TD individuals who tend to exhibit higher Gamma shape values (tending to Gaussian ranges of the Gamma family) in correspondence with lower noise regimes (lower Gamma scale values). These are features indicative of a predictive code whereby events in the future are predictable by present events as much as present events are predictable by past events.

This finding, relating idiopathic ASD with SYNGAP-ASD and CP, may indicate some common etiologies across these conditions as well as a more neurobiological understanding of the inner sensations of patients with chronic pain in terms of cardiac activity. While additional subjects are needed to confirm the observed patterns, such Gamma characterizations of the cardiac signatures proved useful in stratifying atypical vs. typical populations.

### 4.2. Facial Activity

Facial activity and eye movements have recently received support as biomarkers differentiating toddlers who have already received a diagnosis of ASD from those who are developing along a neurotypical trajectory—as far as social interactions and communication go [25,45]. Such studies take a long time to acquire data and the statistical analyses they perform tend to assume theoretical distributions that do away with behavioral nuances as gross data [45,46]. For example, such protocols may require 20–30 min of watching a video or playing a video game at the lab. In our approach, we aimed instead at developing protocols that collect data for a few seconds, under a minute, such that they are brief and yet effective to give us the information that we need to screen for early neurodevelopmental differences in a rather dynamic way.

Under these brief assays of the face, sampling for merely 20 s—inclusive of 5 s practice, 5 s resting, 5 s smiling and 5 s opening the mouth as in a surprised expression—we were able to differentiate in neurotypicals the three regions of the face that are innervated by the trigeminal ganglia (see also [40] extended to 126 participants). Using the micro-movement patterns across these different regions of the face, we also saw significantly different patterns and lack of differentiation across facial regions in the ASD group. Consistently at baseline, the ASD resting patterns are excessively dysregulated and present not only involuntary motions, but also higher levels of random noise derived from the moment-by-moment micro-fluctuations of motion speed-based time series data. When examining the patterns across different subregions of the face, we see in TD participants proper differentiation of facial micro-motions corresponding to the projection of emotional facial features defined by the seven FUMEs. In contrast, in a subset of these FUMEs (happy smile vs. surprise), we see a blend of noisy micro-expressions difficult to discern and partly explained by low to no muscle tone, or spastic micro-motions. Despite their trying to project a facial emotion, to the outsider—exclusively relying on the naked eye—it may appear that the autistic individual is not engaging. Our biometrics clearly show the excessive random noise of the motion patterns at a micro-level, occurring largely beneath awareness and away from naked eye detection ranges. The autistics have a motor control problem [40] rather than a lack of empathy or emotion, as has been previously proposed [47,48].

When assessing facial action units (AUs), we find significantly lower intensities of eyebrow movement in ASD, despite detecting the presence of the AU. This mimics previous work on facial sentiment analysis and emotional engagement in autistics [40,46].

### 4.3. Voice Activity

The analysis of voice data registered during the UPDRS testing revealed significant differences between patients with PD and controls. This result opens the possibility of performing such a portion of clinical tests remotely with commercially available means. In turn, this ability permits broad use of such tests at large scale, inclusive of testing previously underrepresented sectors of the population in our research labs. Voice inherently depends on motor function and proper synergies of the muscles involved in generating it. Vocal acoustic characteristics from cell phones have been previously used to successfully separate patients with PD from controls [49,50] and distinguish, among patients with PD, those with severe freezing gait patterns [51]. Furthermore, using voice analysis in autism has also yielded promising results [52,53].

### 4.4. Motor Activities

Prior work has identified in autism elevated patterns of random noise in the hand speed of pointing behavior [9] and in the saccadic eye movements [25] as well. These results alert us of dysregulation in actions requiring eye–hand coordination, such as those needed to accurately point to visual targets. When exploring the Gamma metrics of a simple reaching activity, we were able to separate TD vs. ASD participants based on their upper arm movement (pointing to a visual target) from video data. In the ASD group, we find that this motor signal exhibits excess variability compared to the TD group. This result supports the notion of disrupted kinesthetic reafference discovered in ASD [9] and is inclusive of disruption of involuntary motions as well [22].

Besides differentiating pointing activities across participants, the present work also distinguishes activities across sensors. These included walking and resting tasks. These patterns were evident in the parameter space spanned by the Gamma NSR derived from the MMS of the IMUs’ acceleration. They were also demonstrated in the center of position for each participant, which differentiated them across tasks for each body part’s location.

These assays were simple to perform and yet very informative of functional capacity for motor control and coordination. They captured signatures and ranges of various stochastic parameters across neurotypical participants and across participants with different neurological and neuropsychiatric conditions. They also distinguished between participants of different age groups. This was possible to do because the MMS are standardized to account for allometric effects due to anatomical disparities across the population, while the Gamma characterization of biorhythmic fluctuations provides interpretable patterns with empirical validation.

## 5. Conclusions

Together, the findings of this study highlight the usefulness of evaluating cardiac, motor, voice, and facial activity when attempting to characterize neurodevelopmental disorders and neurological illnesses such as ASD and PD outside of laboratory settings. The use of easily accessible and commercially available tools and technologies such as smartwatches, microphones, and video cameras along with brief yet informative assays (resting, reaching to a target, smiling, etc.) allow for ease of implementation and training in the comfort of one’s home. Using our standardized biometrics, we find unique patterns in biorhythmic activities that can allow us to differentiate between healthy and disease states as well as characterize a variety of emotional expressions and task types, all under a unifying statistical platform for personalized analysis. The consistencies observed from this study and previous work emphasize how such techniques may be useful for others to implement remote research routines, ultimately allowing such methods to be scalable and translatable to the clinical realm. Future work aims to expand our lab-in-a-box paradigm to include larger and more diverse samples and further assess the effectiveness of applying our analytical approaches.

## 6. Patents

Rutgers University holds copyrights to the “Lab in the Box” concept.

The MMS and Gamma Process approach are used under granted patents by the US Patents and Trademark Office. US10176299B2—Methods for the diagnosis and treatment of neurological disorders; US20170344706A1—Systems and methods for tracking neuro-development disorders; US20170340261A1—System and method for measuring physiologically relevant motion; and US10786192B2—System and method for determining amount of volition in a subject (extended to Germany, France and the UK).

## Figures and Tables

**Figure 1 jpm-15-00463-f001:**
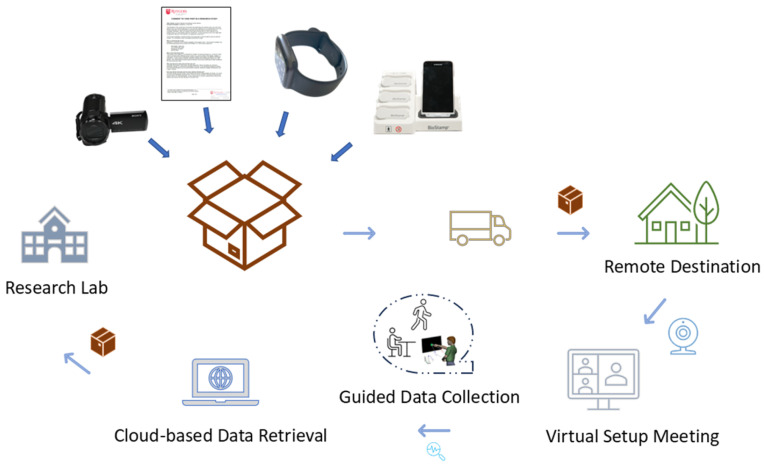
The lab-in-a-box concept and meta-paradigm. Our procedural cyclic approach for data collection outside of laboratory settings is used across all three studies discussed in this work. Study materials (IRB-approved consent forms, instructional manuals) and equipment (video recorder, wearable sensors) are packaged and delivered to a remote destination (home, school, clinic, etc.) for data collection. Virtual meetings are scheduled for setup procedures, guiding, and monitoring the participant through experimental tasks, and the data upload process. Data is then retrieved and assessed for viability remotely at the researcher’s end via HIPAA-complaint cloud-based storage procedures. Once completed, using pre-paid courier services, the items are re-packaged and delivered back to the research lab to then be sent to the next participant.

**Figure 2 jpm-15-00463-f002:**
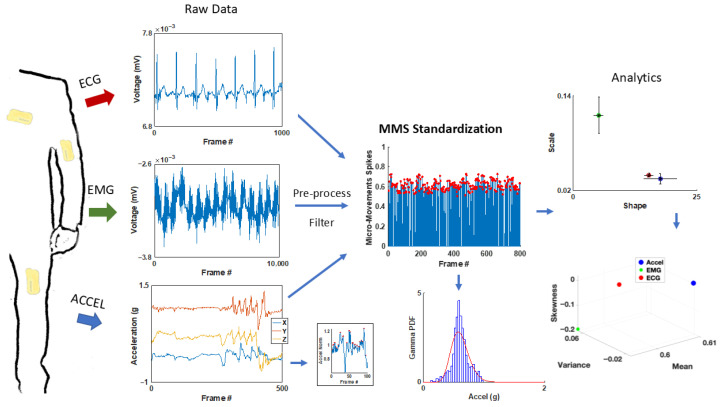
Sensors’ location and analysis pipeline for personalized biometrics. Wearable sensors were placed on the chest (lead II position), specific muscles (such as the bicep), and limbs (such as the anterior thigh), to obtain electrocardiographic (ECG), electromyographic (EMG), and acceleration (ACCEL) data. Each of these biophysical signals were then preprocessed or filtered to obtain the physiologically relevant time series data. The parameters were then normalized in two steps, first by estimating the empirical distribution mean (from a continuous family of probability distribution functions) and then by obtaining the absolute deviations from the empirically estimated mean. The inset panel in the acceleration waveform shows the peaks isolated from the valleys, to locally scale out allometric effects on the acceleration (in this case) from true effects of the task/treatment. Equation (1) in the main text shows the local scaling formula. The micro-movement spikes (MMS) peaks thus obtained reflect individual fluctuations that current methods toss out as gross data (explained in detail here [11]). The MMS peaks were then best fit by the continuous Gamma family of probability density functions (PDF) using maximum likelihood estimation, MLE, with 95% confidence. From the empirically estimated PDF we can then obtain the Gamma shape and scale parameters and the Gamma moments to build parameter spaces and identify those which maximally output treatments’ effects, thus offering personalized targets for treatment. This characterization of each biophysical signal is independent of sampling resolution, functional level of the nervous systems (autonomic, involuntary, reflexive, spontaneous, automatic or voluntary), and functional assay.

**Figure 3 jpm-15-00463-f003:**
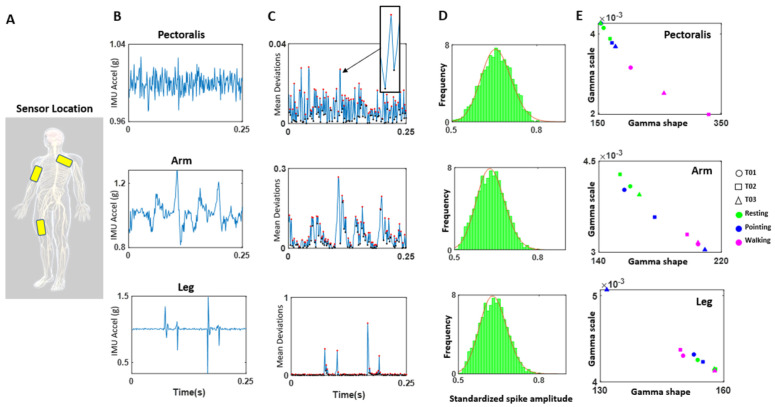
IMUs accelerometer data and pipeline for analysis. (**A**) Sensor locations on pectoralis, arm and leg. (**B**) Sample accelerometer output upon obtaining the Euclidean norm of the triaxial components (1 g is 9.8 m/s^2^). (**C**) Absolute deviations from the empirically derived mean. Red dots signal the peak absolute deviations surrounded by local minima used in the calculation of the MMS (see text). (**D**) Frequency histograms of the peaks from the MMS with best-fitting PDF as the continuous Gamma family with shape and scale parameters. (**E**) Gamma parameters’ plane representing sample data from three participants for each of the resting, pointing, and walking tasks.

**Figure 4 jpm-15-00463-f004:**
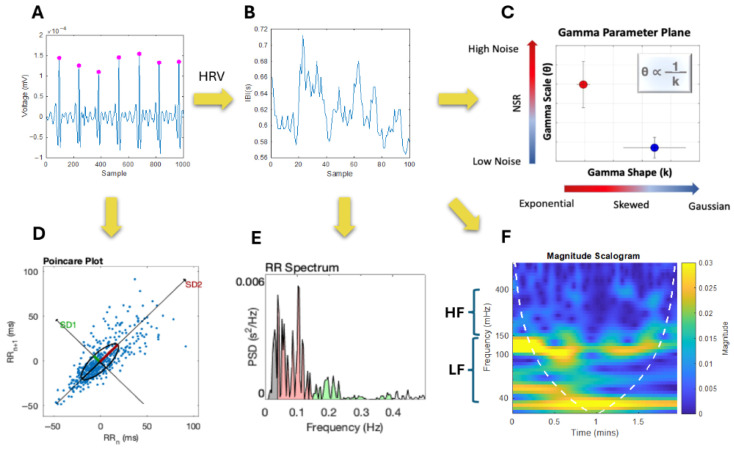
ECG cardiac data and pipeline for analysis. (**A**) R-peaks (highlighted as magenta dots) are detected from the ECG signal used to compute the IBI or RR interval. (**B**) Timing between peaks is used to obtain the IBI series. (**C**) The Gamma scale and shape parameters are calculated from the gamma PDF of the IBI series. (**D**) Time-domain analyses of the IBI series include Poincaré plots that allow us to compute the SD1 and SD2 parameters. (**E**) Frequency domain analyses allows us to compute the frequency power in the LF (red) and HF (green) ranges of the IBI series’ power spectrum. (**F**) Wavelet analyses allow us to visualize frequency changes in the IBI signal across time via visualization of the magnitude scalogram.

**Figure 5 jpm-15-00463-f005:**
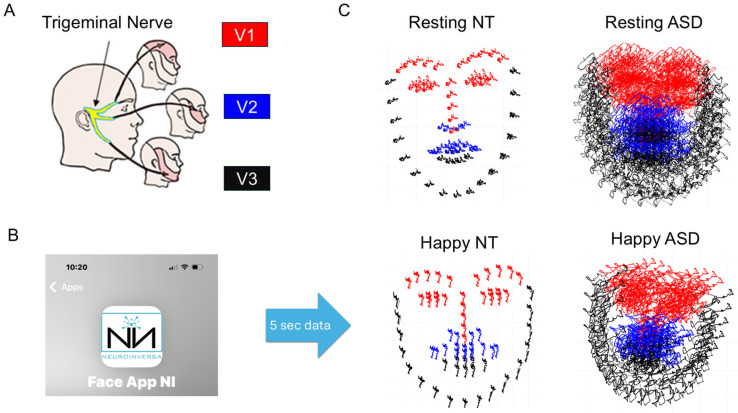
Sample data collected from videos using an app that gathers 5 sec video data. (**A**) The parcellation of the face into three regions V1 (ophthalmic red), V2 (maxillary blue) and V3 (mandibular black), according to the trigeminal nerve innervation of the face. (**B**) Simple app to gather 5 s of data while a person makes a facial expression. (**C**) Sample grid trajectories recovered with Open Face software from resting and happy (smile) in neurotypical (NT) control child and ASD child (higher effort from involuntary micro-movements).

**Figure 6 jpm-15-00463-f006:**
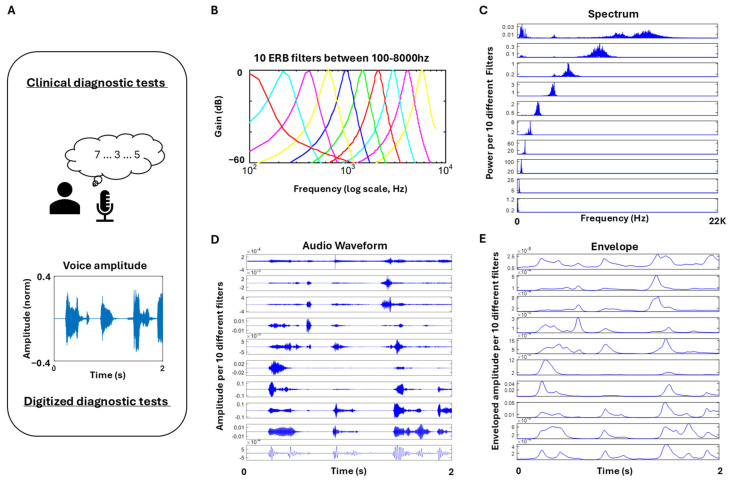
Audio analysis. (**A**) Raw audio data obtained with mic during recollection of digits in memory task. (**B**) The EBR filters (colored curves) between 100 and 8000 Hz used in the decomposition by the *mirfilterbank* function of the MATLAB MIR toolbox (see text). (**C**) Spectrum vs. frequency for each of the 10 filters used. (**D**) Ten independent audio waveform samples and (**E**) their envelope.

**Figure 7 jpm-15-00463-f007:**
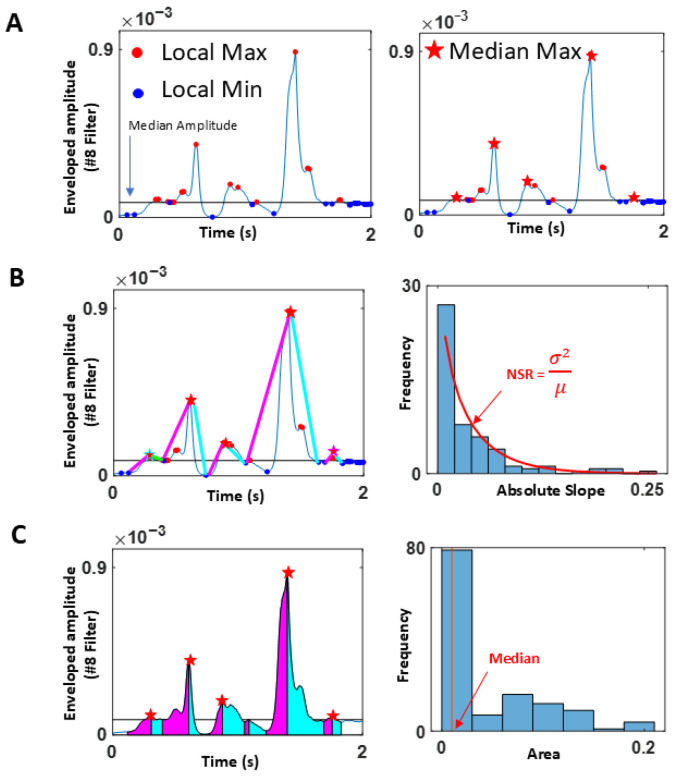
Studying the upward (attack represented in magenta) and downward (decay represented in cyan) phases of the audio waveform. (**A**) The enveloped amplitude from the 8th filter was used to obtain the waveform featuring multiple local peaks and valleys. The median amplitude value was obtained to automatically threshold the maxima and minima above and below it, respectively. The absolute maxima among multiple consecutive local maxima were then used to define the maxima of interest and unambiguously distinguish it from the surrounding local minima. (**B**) The slopes of the upward (attack) and downward (decay) phases were thus defined to study the moment-by-moment stochastic variations and to empirically characterize them using the Gamma noise-to-signal ratio (Gamma scale parameter empirically estimated using MLE with 95% confidence intervals). (**C**) The area under the curve is also quantified along with its variability relative to the median value.

**Figure 8 jpm-15-00463-f008:**
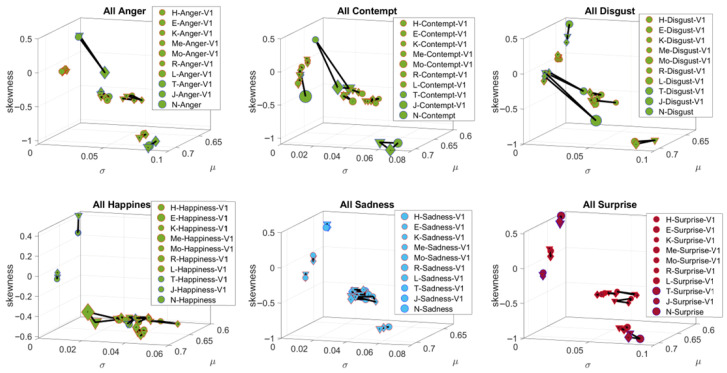
Characterization of six of the seven FUMEs for visualization purposes, across 10 individuals. These signatures automatically separate participants into different subregions of the parameter space spanned by the Gamma mean (x-axis), Gamma variance (y-axis), Gamma skewness (z-axis), and Gamma kurtosis (proportional to the size of the marker, with larger size representing more kurtotic distributions). Vertices of the triangle are in correspondence with the three facial regions innervated by the trigeminal nerves, V1, V2 and V3 (see main text, V1—ophthalmic, V2—maxillary, V3—mandibular). Some participant triangles cluster in different subregions, while some participants span larger triangles for some emotions.

**Figure 9 jpm-15-00463-f009:**
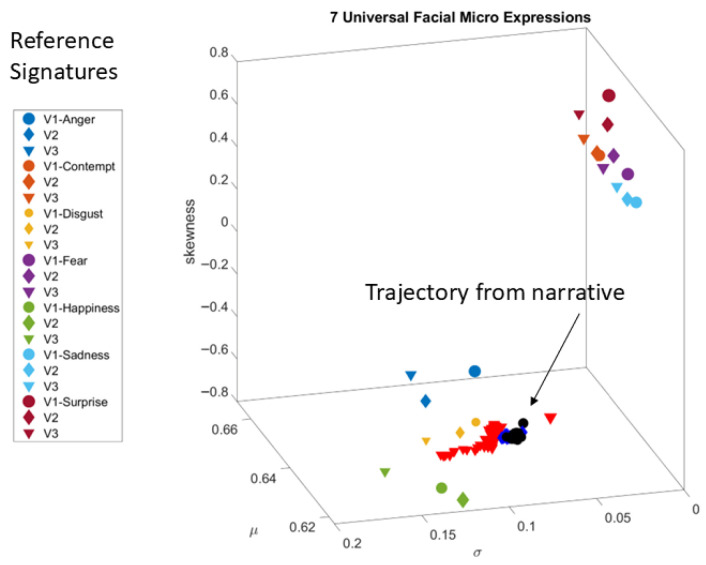
Gamma parameter space representation of a narrative from video (sentiment) analyses, where the face positional trajectories from each region V1, V2, V3 are unfolded frame by frame, relative to the signatures of the seven FUMEs of a model participant. Gamma stochastic signatures from the micro-movements’ trajectories spanned by the V1–V2–V3 regions of the face are plotted in the order in which they are obtained, one frame at a time, with a frame corresponding to 5 s of activity and using 50% overlap between windows. The markers used to represent the narrative trajectory are as in Figure 7. The red triangles are from the V1 (ophthalmic) region and these span the largest excursions. The blue diamonds are from the V2 (maxillary) nose region and the black circles are from the V3 (mandibular) region. They localize closest to the Disgust region of the reference model, farthest from surprise, sadness and contempt.

**Figure 10 jpm-15-00463-f010:**
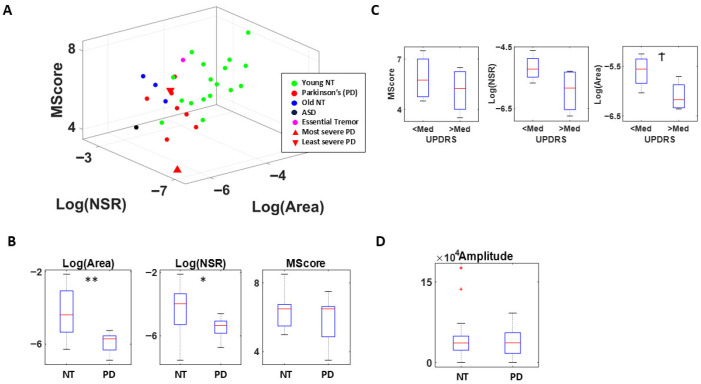
Voice analysis revealed differences between the patients with PD and controls. (**A**) Parameter space to localize patients with PD, age-matched controls, and young neurotypical controls according to the log(Gamma NSR) along the x-axis, the log(area under the curve defined by the slopes), and the memory score from remembering the digits being said and recorded with the microphone. (**B**) Statistical differences between NT and PD participants according to the non-parametric Kruskal–Wallis test (* *p* < 0.05, ** *p* < 0.01). (**C**) Differences in UPDRS scores above and below the median (outliers marked in red asterisks), cross indicates trends non-significant at *p* < 0.1. (**D**) Differences in amplitude fluctuations between patients with PD and age-matched controls.

**Figure 11 jpm-15-00463-f011:**
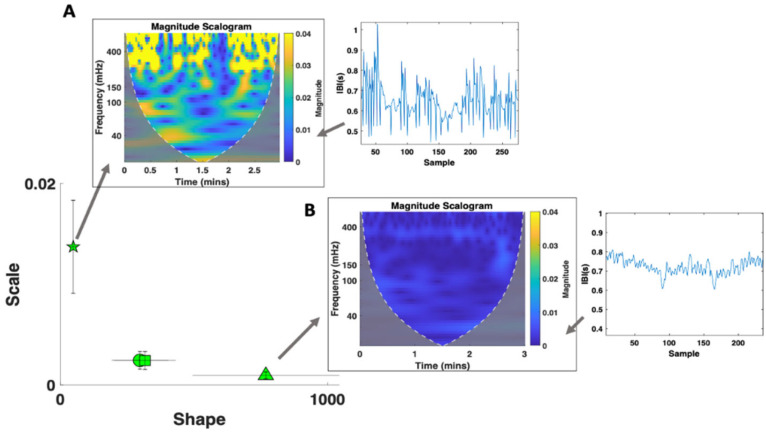
Characterization of the Frequency and Stochastic signatures featured in the HRV parameters and MMS-Gamma analyses (shown here is the resting task for four different subjects). (**A**) Higher scale (NSR) and lower shape values correspond with an elevated randomness in the IBI signal. This is also characterized by atypical ranges in the magnitude scalogram depicting the low frequency (LF) and/or the high frequency HF ranges of the magnitude scalogram. This higher random noise (low predictability of the signal) is often observed in anxiety states and is apparent in the IBI series of the subject. (**B**) Lower scale (NSR) and higher shape tending towards symmetric Gaussian ranges of the Gamma family are associated with a more balanced magnitude scalogram and greater predictability of the IBI series, often observed in the responses of neurotypical signals.

**Figure 12 jpm-15-00463-f012:**
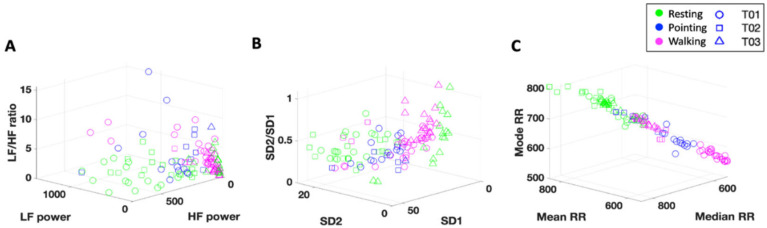
Comparison of HRV parameters across the resting (green), pointing (blue), and walking (magenta) tasks for all three subjects. (**A**) In the frequency domain, we see that the resting task exhibits a wider range of values within the LF and HF range compared to the other tasks; however, we see higher LF/HF ratios for both the pointing and walking tasks. (**B**) In the time domain, we again see a greater spread between the SD1 and SD2 parameters for the resting task. The walking task exhibits an overall lower SD1 accompanied by a higher SD2/SD1 ratio compared to the pointing task. (**C**) When observing the Gaussian moments of the IBI series, we see relatively clearer separation between the tasks, with the resting task exhibiting the highest moment values (which corresponds to a lower heart rate), the walking task exhibiting the lowest (highest heart rate), and the pointing task sitting in between. Such HRV findings are consistent in characterizing the physical exertion associated with each task.

**Figure 13 jpm-15-00463-f013:**
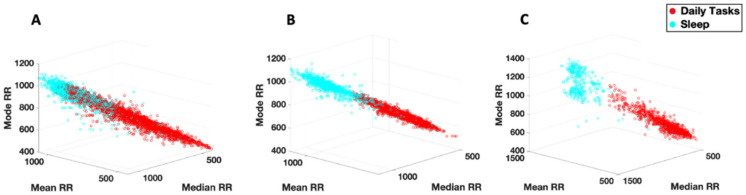
Separating daily activities from sleep. Gaussian moments of IBI series for each subject performing Daily and Sleep tasks. All subjects demonstrate decreased moments during sleep compared to daily miscellaneous activities. (**A**) Subject T01 demonstrates high overlap between the two tasks whereas subject T02 (**B**) shows moderate overlap. Subject T03 (**C**) exhibits the greatest separation/least overlap between sleep and daily activities.

**Figure 14 jpm-15-00463-f014:**
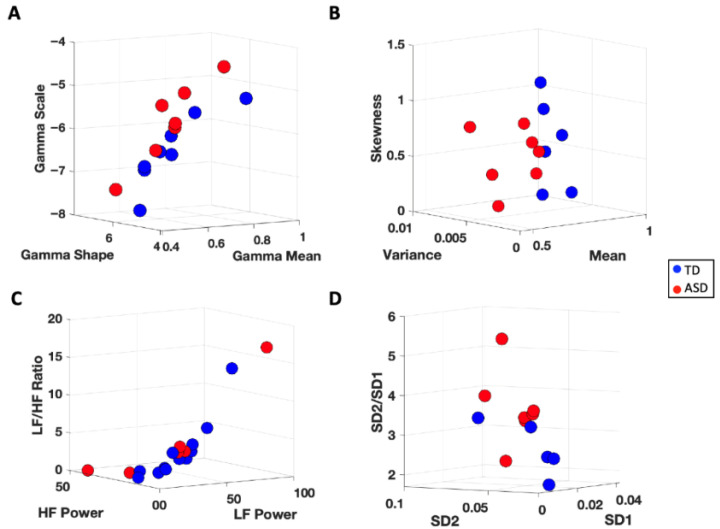
Comparison of signatures of cardiac activity between TD and ASD participants performing the Walking task. (**A**) From the Gamma distribution, the cardiac activity of ASD subjects tends to exhibit higher Gamma scale values (higher NSR) compared to TD subjects. (**B**) From the Gaussian distribution of the IBI series, ASD participants show higher variance. (**C**) Frequency domain parameters do not demonstrate as clear a separation between ASD and TD subjects. (**D**) In the time domain, the cardiac activity of ASD subjects tends to exhibit a higher SD2 and SD2/SD1 ratio compared to TD subjects, suggesting sympathetic overdrive in autism.

**Figure 15 jpm-15-00463-f015:**
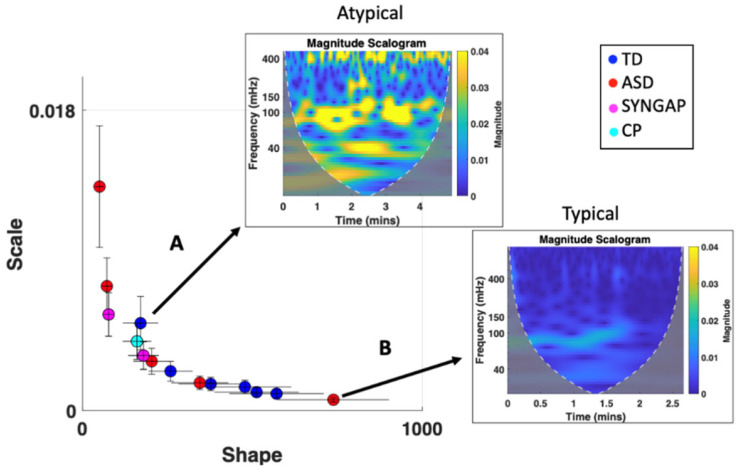
Gamma parametric characterization of cardiac activity for TD and ASD participants including those with SYNGAP and chronic pain (CP). ASD and SYNGAP subjects tend to exhibit higher scale and lower shape parameters compared to TD subjects. The case of the chronic pain (CP) subject (cyan) also falls in a similar region of the Gamma parameter plane. (**A**) Case of TD subject that exhibits atypical cardiac activity with high magnitude in the LF range. (**B**) Case of ASD subject with a relatively typical cardiac pattern. Such analyses can help characterize states of dysregulation while stratifying participants based on the signatures of their cardiac activity.

**Figure 16 jpm-15-00463-f016:**
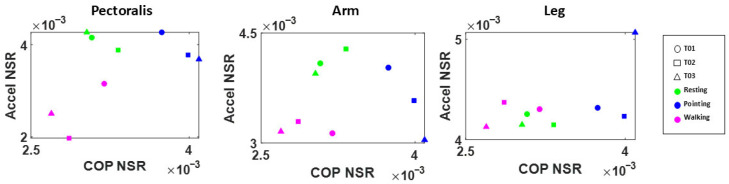
Differentiating activities across sensor locations and participants. A parameter space spanned by the Gamma NSR derived from the MMS of the IMUs’ acceleration and the center of position for each participant differentiates across tasks for each body location (pectoralis, arm, and leg).

**Figure 17 jpm-15-00463-f017:**
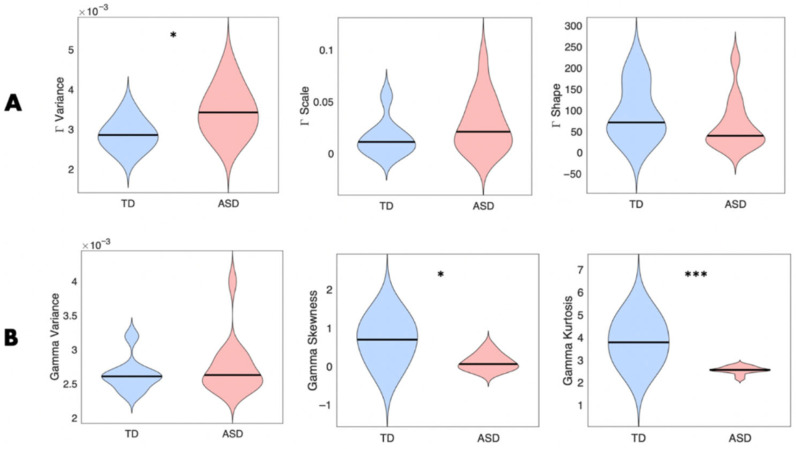
Gamma parameters characterizing IMU acceleration MMS for TD vs. ASD participants during the Pointing Task. (**A**) Significantly higher Gamma variance (median and range) for the upper arm in the ASD group vs. the TD group. This is associated with elevated Gamma scale and lower Gamma shape in the ASD group. (**B**) For movement of the chest/pectoral area, we observe significant differences in the Gamma skewness and kurtosis between TD and ASD groups. Differences in the Gamma variance for pectoral movement were not significant across groups. * *p* < 0.05, *** *p* < 0.001.

**Figure 18 jpm-15-00463-f018:**
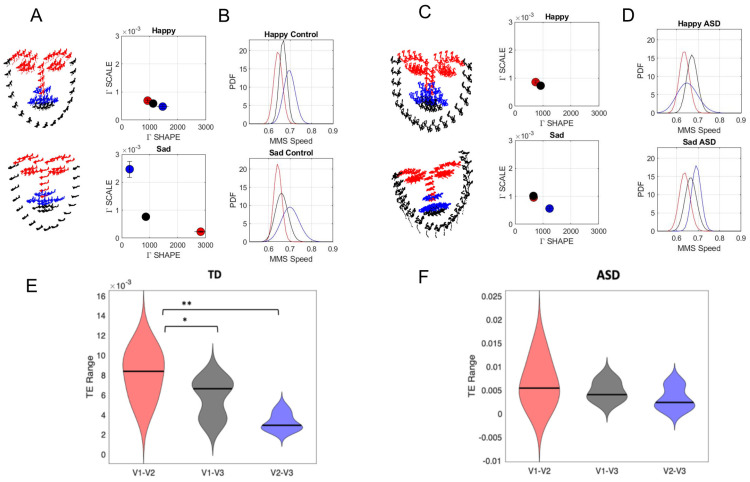
Differentiation between Controls and ASD facial micro-movements. (**A**) Facial grid trajectories from typically developing TD control child in happy vs. sad micro-expressions expressed across three regions of the face (ophthalmic V1 color coded in red, maxillary V2 color coded in blue and mandibular V3 color coded in black). (**B**) The empirically estimated Gamma parameters correspond to the three regions in (**A**) for the TD child plotted on the Gamma parameter plane with 95% confidence intervals along the shape and scale (noise to signal ratio) axes. Corresponding Gamma probability density functions distinguish the happy vs. sad micro-expressions. (**C**) Facial grid trajectories for the ASD child color coded as in (**A**). (**D**) Gamma plane and Gamma PDFs as in (**B**) for the ASD child. (**E**) Pairwise Transfer Entropy TE differentiates facial synergies in TD. Yet, this differentiation is lost in ASD (**E**) according to the group analysis. Range of transfer entropy values for V1→V2 is significantly wider compared to the TE range for V1→V3 and V2→V3 for TD subjects. Range of transfer entropy values across V1, V2, and V3 regions is relatively the same for ASD subjects, * *p* < 0.05, ** *p* < 0.01. (**F**) Pairwise Transfer Entropy TE fails to differentiates facial synergies in ASD.

**Table 1 jpm-15-00463-t001:** Summary of the three studies.

Study Name	Sample Size	Age|Sex	Diagnoses	Data Type(s) Collected
FUME Assessment	10	28 +/− 5 (3 M, 7 F)	10 TD	Face ^+^
PD Evaluation	30	59.8 +/− 10.7 (22 M, 8 F)	20 TD 8 PD 1 ET, 1ASD	Voice ^++^
Dysregulation Screening	18	20.2 +/− 12.10 (9 M, 9F)	10 TD 8 ASD	Face ^+^, Motor ^+++^ Heart ^++++^

^+^ Video (face) data in f/s, ^++^ Microphone (Voice) data unitless, ^+++^ Accelerometer inertial measurement units IMUs (Motor) data in m/s^2^, ^++++^ (electro-cardiography ECG (Heart) in mV/s), FUME stands for Facial Universal Micro Expressions, PD stands for Parkinson’s Disease, TD stands for Typically Developing, ASD stands for Autism Spectrum Disorders, ET stands for Essential Tremor.

## Data Availability

The original contributions presented in this study are included in the article. Further inquiries can be directed to the corresponding author.

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
