# Peer review of "Setting Up Our Lab-in-a-Box: Paving the Road Towards Remote Data Collection for Scalable Personalized Biometrics"

_jpm, 2025, doi:10.3390/jpm15100463_

Round 1
Reviewer 1 Report
Comments and Suggestions for Authors
The manuscript presents a novel and timely approach to decentralized behavioral and physiological data collection through a remote, multi-modal sensing system. The proposed framework integrates wearable devices, video capture, and voice analysis to enable remote assessment of neurotypical and atypical populations, including individuals with Parkinson’s disease and autism spectrum disorders. The authors further apply a unified statistical framework (Micro-Movement Spikes and Gamma distribution modeling) to derive individualized biomarkers from ECG, IMU, and facial activity data. From the reviewer’s perspective, this manuscript should address the following concerns:
- While the manuscript offers a wealth of information, it is at times verbose and could benefit from more focused organization. Particularly, the Introduction should be strengthened by a clearer articulation of the knowledge gap, specific research objectives, and the rationale behind the three studies.
- The manuscript presents a range of novel metrics but lacks systematic statistical comparison across groups in several instances. Greater use of effect sizes, confidence intervals, and p-values within figures would enhance interpretability and scientific rigor.
- The relatively small number of participants limits the generalizability of the findings. The authors are encouraged to acknowledge this limitation more explicitly and outline plans for scaling up in future work.
- Several figures are dense and may benefit from simplification or segmentation. Enhanced use of color-coding, statistical overlays, and legends would improve clarity.
-
While some self-citations are reasonable, several other references appear unnecessary and can be removed: 5, 8, 16–18, 20–23, 39–46, 50, and 61. These citations are:
Associated with content that could be trimmed from the overly long introduction,
Not explicitly cited or discussed in the main text,
Only loosely related to the study’s findings, or
Redundant with other, more relevant citations.
Comments on the Quality of English Language
From the reviewer’s perspective, this manuscript can be accepted after addressing the following concerns:
Author Response
Thank you for your review.
The manuscript presents a novel and timely approach to decentralized behavioral and physiological data collection through a remote, multi-modal sensing system. The proposed framework integrates wearable devices, video capture, and voice analysis to enable remote assessment of neurotypical and atypical populations, including individuals with Parkinson’s disease and autism spectrum disorders. The authors further apply a unified statistical framework (Micro-Movement Spikes and Gamma distribution modeling) to derive individualized biomarkers from ECG, IMU, and facial activity data. From the reviewer’s perspective, this manuscript should address the following concerns:
1. While the manuscript offers a wealth of information, it is at times verbose and could benefit from more focused organization. Particularly, the Introduction should be strengthened by a clearer articulation of the knowledge gap, specific research objectives, and the rationale behind the three studies.
We thank the reviewer for the opportunity to improve our paper. We have now rewritten the introduction and re-edited the entire paper to improve clarity.
2. The manuscript presents a range of novel metrics but lacks systematic statistical comparison across groups in several instances. Greater use of effect sizes, confidence intervals, and p-values within figures would enhance interpretability and scientific rigor.
The methods used in the paper are a new generation of statistical techniques invented in our laboratory and patented in the US and the EU. These techniques inherently include the use of continuous families of probability distributions that are empirically estimated. The points reported are precisely the parameters unfolding such distribution families and each comprises hundreds of data measurements. The use of p-values and traditional metrics of comparisons between conditions/groups under the significant hypothesis testing model does not apply to this type of data outcome. Those traditional metrics are based on single theoretical distributions such as the Gaussian and call instead for measurements regarding significant shifts in mean and variance (the assumed moments of the theoretically assumed distribution describing the stochastic process). The data of interest here (time series of hundreds of values of a multiplicity of parameters, does not distribute in any symmetric way and cannot be subject to such theoretical assumptions. We have (when appropriate) used non-parametric comparisons (e.g., Kruskal-Wallis test) to show differences in probability distribution parameters. We note here that we are not working
with the raw data for these comparisons as the raw data comprises families of distributions which shifting parameter ranges in non-stationary ways. Furthermore, since our parameter spaces are distribution based, we cannot use distance metrics such as the Euclidean to measure differences. Instead, we measure distance in spaces spanned by parameters of probability distributions. Our points in the parameter spaces of interest are probability distributions and as such we use metrics such as the Wasserstein’s distance metric (aka as the Earth Movers’ Distance in ML/AI fields)
Whenever possible, we report the p-values from non-parametric comparisons (e.g., in Figure 16 and lines 767 and 774. We also evaluate the Transfer Entropy pairwise between regions and report p-values signaling statistical differences at significant levels in line 802 and figure 17.
3. The relatively small number of participants limits the generalizability of the findings. The authors are encouraged to acknowledge this limitation more explicitly and outline plans for scaling up in future work.
This relates to the new techniques that we have created which extract thousands of measurements from each participant. As such, each participant can be conceptualized as a case study whereby the statistical power relies on the number of measurements per participant, rather than on the number of participants. Each participants spans a continuous family of probability distributions and encompasses unique signatures amenable to use in personalized medicine – the topic of this journal. The significance of this approach is that instead of working with population statistics derived from assumed theoretical distributions, we work with the person’s individualized signatures derived from the inherent fluctuations of the person, which are empirically estimated with 95% confidence, using maximum likelihood estimation. Whereas traditional approaches employ a one-size-fits-all model and impose a theoretical distribution on the fluctuations of the data parameters, ours estimate the distribution(s) family that best fits the parameter’s fluctuations. Such fluctuations have also been standardized to account for allometric effects due to anatomical size disparities across subjects. As such, we can place all subjects’ derived distribution parameters on the same parameter space and establish (1) comparisons, (2) classification, (3) self-emerging clusters that automatically group participants in different categories.
The modest size of the various cohorts that we present are merely samples to provide methodological information and descriptions of our data analysis pipelines. To provide large numbers of subjects we refer to our prior literature featuring thousands of individuals. We have cited this literature from our lab throughout the text to provide the validity of our methods, already established over the span of more than 2 decades of work. The present paper is merely a proposition of a new concept to scale the research using these methods in combination with wearables and new digital advances accessible to most people. Our methods make it possible to do so independently of sensor type, body part, task assay or sampling resolution.
We have added these comments and explanations in the Methods and Discussion Section now.
4.Several figures are dense and may benefit from simplification or segmentation. Enhanced use of color-coding, statistical overlays, and legends would improve clarity.
We have added such features to the figures.
5.While some self-citations are reasonable, several other references appear unnecessary and can be removed: 5, 8, 16–18, 20–23, 39–46, 50, and 61. These citations are: Associated with content that could be trimmed from the overly long introduction, Not explicitly cited or discussed in the main text, Only loosely related to the study’s findings, or Redundant with other, more relevant citations.
We appreciate the suggestion of the reviewer. We now explain the citations better and relate them directly to the content of the present work. Those which were not needed were eliminated and new ones from other labs doing similar work were added. We note however that inevitably, given the novelty and uniqueness of these methods, we must cite the prior work that the lab has done to validate these methods across thousands of individuals of different ages and across disorders of the nervous system.
Reviewer 2 Report
Comments and Suggestions for Authors
This manuscript presents a novel approach to remotely characterize behavioral states using multimodal biophysical data (cardiac signals, motor kinematics, voice, and facial activity), collected in naturalistic, non-laboratory settings. Using commercially available devices such as smartwatches, microphones, and cameras, the study demonstrates accessible methods for assessing neurodevelopmental and neurodegenerative disorders—specifically autism spectrum disorder and Parkinson’s disease. By applying various techniques to these diverse data streams, the authors identify distinct biorhythmic patterns that differentiate healthy individuals from those with neurological conditions and characterize various emotional expressions and task types. This work emphasizes the potential for personalized biometrics that can be implemented outside traditional laboratory environments, facilitating clinical translation and broader applications in remote healthcare, education, and research.
While the overall concept of the study is promising, the presentation of the details and results is overly complex and difficult to follow. To aid clarity and enhance understanding, I have outlined several major revision remarks that require attention, with the aim of improving the accessibility and coherence of the methods and results within this extensive work.
Detailed revision remarks:
- The title: The term “scalable” is used, however, its interpretation is not explained in all relevant sections: Abstract, Methods and Results.
- The Abstract must be fully rewritten with the following considerations:
- Background: “With the rise of the COVID-19 pandemic…” -> The aims of the study are not well expressed. Thy must stress novel but not past problems, novel developments and the expected future benefits. Emphasis should be placed on what is new, relevant, and forward-looking in the context of the study.
- Methods: This sub-section does not reveal sufficient technical details for the developments in this study. Avoid generalized or vague descriptions that could apply broadly to other studies in the field. Instead, provide precise and study-specific information, including the techniques used, the types of signals analyzed, and the processing methods employed. This will enhance the clarity, reproducibility, and scientific value of the work. .
- Methods: The section should include clear and detailed information about the data collected in this study, such as the source, size, characteristics, and any relevant acquisition parameters.
- Results: Numerical results are missing. Add important numerical results that that support the study’s main hypothesis and demonstrate the effectiveness or relevance of the proposed approach.
- Conclusions: Not supported by the results. Ensure that all claims are directly grounded in the data and analysis provided in the results section.
- Introduction: The introduction contains long paragraphs without references, which is not acceptable in a scientific context. One of the main purposes of Introduction is to establish the foundation of existing knowledge by referencing relevant literature. Introduction includes only 8 references, which are super insufficient for a well-established and widely studied field such as human biometrics based on physiological signals. For example, the authors need to reference studies using ECG and motor (kinematic or planar) signals for biometrics, such as but not limited to: [https://doi.org/10.1016/j.patcog.2013.06.028, https://www.sciencedirect.com/science/article/pii/S1566253519302039#bib0038, https://doi.org/10.3390/s18020372, https://doi.org/10.1038/s41597-023-02077-3, etc.].
- Follow the journal template – All subsections (and equations) must follow the defined style for numbering format. Figures and tables must appear in the first paragraph right after their first mention in the text.
- Ln 119: Define the date of the official approval protocol.
- Figure 1: Not enough informative for the sources of information, data flow and processing steps. The figure must be clarified with sufficient text or other details, which clarify the methods and follow the flow of information in section Materials and Methods. As this is the main block diagram of the study, I miss essential information regarding the defined Study1, Study2 and Study 3 procedures and clear description of sequential signal processing steps. In my opinion, the figure must be substantially improved.
- Table 1: (1) The reference number of the studies (official registry) must be added; (2) Abbreviations must be defined; (3) Measurement units must be added; (4) The last column is not informative to specific “Data types” because “data” is usually connected to specific physiologic signals or specific measurements. Define the type of the recorded signals.
- Figure 2 (Pg.4) contains excessive detail and lacks clarity. While it is intended to be a methodological overview, it includes too much information related to the results, which detracts from its primary purpose. The figure should be significantly simplified to focus on key elements such as the placement of biosensors on the human body and clearly labeled blocks that follow exactly to the signal processing procedures outlined in Section 2: Materials and Methods. Its goal should be to provide a high-level overview of the processing pipeline, without delving into low-level technical details—those should be reserved for separate, dedicated figures. Final note: always check the font size and element details if readable in the pdf file.
- Ln 142 contradicts Ln 119: There is a lack of clarity regarding the ethical approvals referenced in the manuscript. It is confusing that different approvals from the same institution appear to be associated with the same study. The author must explicitly specify how many distinct approvals were obtained, which ethical committees granted them (number, dates), and how these approvals correspond to each sub-study within the overall research. This information is essential to ensure transparency and compliance with ethical standards.
- Section “Materials & Equipment” must be clarified. The position and type of biosensors is unknown. The type and version of the hardware/software equipment is also unknown. This section must be totally rewritten.
- Ln 184: „standard difference” -> Is this standard deviation? Difference can be mislead as difference of means or other statistical measures. Check for other discrepancies.
- Ln 202: “biorhythms along different systems” -> The context and setting of this phrase need to be clarified. The current description is overly narrative and lacks sufficient technical grounding. The entire section describing Study 3 should be restructured into a clear, step-by-step format, ensuring that each methodological stage is explicitly detailed and easy to follow.
- Confusing that Figure 2 is present in Pg. 4 but also in Pg. 10. This duplication makes questionable whether the correct number format is mentioned in the text, such as in the section “Motor Activity (Wearable Sensors Accelerometer Data)”.
- Figure 2 (Pg.10) -> Measurement units are missing. The font is very small and texts is unreadable. Correct this discrepancy in all other figures in methodological section (Figures 2 to 6).
- The signal processing steps illustrated in Figures 2 to 6 are overloaded with information and presented in a highly narrative style, making it difficult to clearly identify the key output parameters computed for each signal across Studies 1, 2, and 3. To improve clarity, the manuscript should include a dedicated summary section within the Methods—such as a “Statistical Measures” subsection—that explicitly defines all measured parameters at each processing stage and outlines the statistical techniques used to analyze them. Additionally, it is essential to clearly specify the output classes for each study and explain how these classes are standardized across the different studies to enable consistent evaluation and comparison of results.
- Figures 7 and 8 present 3D visualizations of the mean, standard deviation, and skewness for three facial nodes labeled V1, V2, and V3. However, the Methods section does not provide sufficient detail about these nodes or clearly specify the exact measurements they represent. A more thorough description of these facial nodes and their corresponding data should be included to ensure the figures are fully interpretable.
- Result section does not present any kind of statistical comparison between groups with reporting respective p-value estimations, which can prove that the computed metrics can be effectively used for personalized biometrics.
- The results rely heavily on graphical presentations and lack direct comparability between studies. To facilitate evaluation and comparison, a summary table should be included that reports the means or medians along with standard deviations or interquartile ranges for all computed biometric measures. Additionally, this table should present inter-group statistical comparisons to clearly highlight significant differences across the studies (according to the previous comment).
- The Introduction section of the manuscript currently contains a very limited number of references (only 8), which is insufficient for the broad and well-established field of biosignal processing applied to biometrics. Furthermore, the manuscript relies heavily on the authors’ own work, with 23 out of 61 references (38%) being self-citations. The authors should expand their literature review to include a wider range of relevant studies
Author Response
Thank you for your review.
This manuscript presents a novel approach to remotely characterize behavioral states using multimodal biophysical data (cardiac signals, motor kinematics, voice, and facial activity), collected in naturalistic, non-laboratory settings. Using commercially available devices such as smartwatches, microphones, and cameras, the study demonstrates accessible methods for assessing neurodevelopmental and neurodegenerative disorders—specifically autism spectrum disorder and Parkinson’s disease. By applying various techniques to these diverse data streams, the authors identify distinct biorhythmic patterns that differentiate healthy individuals from those with neurological conditions and characterize various emotional expressions and task types. This work emphasizes the potential for personalized biometrics that can be implemented outside traditional laboratory environments, facilitating clinical translation and broader applications in remote healthcare, education, and research.
While the overall concept of the study is promising, the presentation of the details and results is overly complex and difficult to follow. To aid clarity and enhance understanding, I have outlined several major revision remarks that require attention, with the aim of improving the accessibility and coherence of the methods and results within this extensive work.
Detailed revision remarks:
1. The title: The term “scalable” is used, however, its interpretation is not explained in all relevant sections: Abstract, Methods and Results.
We thank the reviewer for taking the time to read our paper and to provide valuable feedback. We have now addressed the term scalable in all aspects of the abstract and across the paper and better motivated their use.
We emphasize the fact that we use commercially available tools of low cost, which we have validated over the span of two decades against high-grade research sensors. We have also simplified the assays to be brief and simple and to allow for frequent measurements and reporting. All algorithms described are embeddable in wearables and have been already embedded through operational apps in smart phones and tablets.
2. The Abstract must be fully rewritten with the following considerations:
•Background: “With the rise of the COVID-19 pandemic…” -> The aims of the study are not well expressed. Thy must stress novel but not past problems, novel developments and the expected future benefits. Emphasis should be placed on what is new, relevant, and forward-looking in the context of the study.
We have rewritten the abstract and the introductions. We emphasize the need for personalized and standardized digital biometrics that handle the heterogeneity in anatomical disparities across ages and rates of neurodevelopment. The methods are agnostic of the modality of data, sampling resolutions and type of biorhythm. As such they can be combined on parameter spaces revealing self-emerging clusters. We discuss these in the intro and across the methods section.
•Methods: This sub-section does not reveal sufficient technical details for the developments in this study. Avoid generalized or vague descriptions that could apply broadly to other studies in the field. Instead, provide precise and study-specific information, including the techniques used, the types of signals analyzed, and the processing methods employed. This will enhance the clarity, reproducibility, and scientific value of the work.
We have rewritten the methods to reflect the contributions of these methods to decentralizing data collection and extending it from lab settings to the comfort of the home environment.
•Methods: The section should include clear and detailed information about the data collected in this study, such as the source, size, characteristics, and any relevant acquisition parameters.
We have now devoted great length to explaining procedures and methods, size, characteristic, and relevant acquisition of various parameters, explaining their standardization as well. Table 1 summarizes the demographic info on all three studies. While Figure 1 explains at a metalevel the concept of the lab in the box, Figures 2.1 and 2.2 and 3-6 explain all procedural details and instrumentation of choice. The methods are general and function for any biosensor or video-based pose estimation. These are further detailed under Procedures starting on line 206 to line 636.
•Results: Numerical results are missing. Add important numerical results that that support the study’s main hypothesis and demonstrate the effectiveness or relevance of the proposed approach.
The study does not have a hypothesis. This is a methodological paper to show how to implement a concept of bringing the research from our labs to the participant’s home. We have detailed all procedures and numerical results, tabulating them across figures 7-8 for the face, with results from non-parametric Kruskal Wallis test in figure 9 and reported p-values across the results. Figure 9A shows the parameter space with points representing distributions that differ across each case study participant. Each point in the scatter is a direct measurement comprising thousands of measurements that span families of distributions. The non-parametric tests are on distribution parameters and reveal self-emerging clusters across the heterogeneous cohort. This could not be done with a one-size-fits-all approach that traditional methods impose on data that is not normally
distributed and violates assumptions of stationarity and linearity of the models a priori imposed on such data.
•Conclusions: Not supported by the results. Ensure that all claims are directly grounded in the data and analysis provided in the results section.
All conclusions are now grounded in the results and explained throughout the methods.
3.Introduction: The introduction contains long paragraphs without references, which is not acceptable in a scientific context. One of the main purposes of Introduction is to establish the foundation of existing knowledge by referencing relevant literature. Introduction includes only 8 references, which are super insufficient for a well-established and widely studied field such as human biometrics based on physiological signals. For example, the authors need to reference studies using ECG and motor (kinematic or planar) signals for biometrics, such as but not limited to: [https://doi.org/10.1016/j.patcog.2013.06.028, https://www.sciencedirect.com/science/article/pii/S1566253519302039#bib0038, https://doi.org/10.3390/s18020372, https://doi.org/10.1038/s41597-023-02077-3, etc.].
We have rewritten the introduction and documented all references from other labs and from our own lab.
4.Follow the journal template – All subsections (and equations) must follow the defined style for numbering format. Figures and tables must appear in the first paragraph right after their first mention in the text.
Adjustments made accordingly
5. Ln 119: Define the date of the official approval protocol.
Date has been added.
6. Figure 1: Not enough informative for the sources of information, data flow and processing steps. The figure must be clarified with sufficient text or other details, which clarify the methods and follow the flow of information in section Materials and Methods. As this is the main block diagram of the study, I miss essential information regarding the defined Study1, Study2 and Study 3 procedures and clear description of sequential signal processing steps. In my opinion, the figure must be substantially improved.
The figure is now better explained clarifying that this is a meta paradigm to show schematically the steps that each participating family took in the cases where the research was remotely performed using our lab in the box concept and recycling the sensors and protocols training from one family to the next.
7. Table 1: (1) The reference number of the studies (official registry) must be added;
These have been added in line 152 before table 1 and line 180
(2) Abbreviations must be defined; (3) Measurement units must be added;
Please see the table caption now with all details added
(4) The last column is not informative to specific “Data types” because “data” is usually connected to specific physiologic signals or specific measurements. Define the type of the recorded signals.
The data type here refers to face, voice, motor and ECG with physical units now included in the caption.
8. Figure 2 (Pg.4) contains excessive detail and lacks clarity. While it is intended to be a methodological overview, it includes too much information related to the results, which detracts from its primary purpose. The figure should be significantly simplified to focus on key elements such as the placement of biosensors on the human body
These have been clearly explained and highlighted in yellow on the heart, bicep and thigh
9. and clearly labeled blocks that follow exactly to the signal processing procedures outlined in Section 2: Materials and Methods. Its goal should be to provide a high-level overview of the processing pipeline, without delving into low-level technical details—those should be reserved for separate, dedicated figures.
The pipeline for signal processing has been simplified and illustrated for one case so we expand to the cohort in the results section.
10. Final note: always check the font size and element details if readable in the pdf file.
We have made sure that the pdf fonts are visible.
11. Ln 142 contradicts Ln 119:
There is a lack of clarity regarding the ethical approvals referenced in the manuscript. It is confusing that different approvals from the same institution appear to be associated with the same study.
We clarified that this is the same protocol and explicitly stated the date and type of protocol.
The author must explicitly specify how many distinct approvals were obtained, which ethical committees granted them (number, dates), and how these approvals correspond to each sub-study within the overall research. This information is essential to ensure transparency and compliance with ethical standards.
This is explained in line 180 – “We note here that this is a blanket IRB protocol to work with digital data and unconstrained motions occurring at the lab, and/or remotely at home, schools and clinics.”
12. Section “Materials & Equipment” must be clarified. The position and type of biosensors is unknown. The type and version of the hardware/software equipment is also unknown. This section must be totally rewritten.
Section clarified with system details and sensor types/positions.
13. Ln 184: „standard difference” -> Is this standard deviation? Difference can be mislead as difference of means or other statistical measures. Check for other discrepancies.
Addressed as standard deviation
14. Ln 202: “biorhythms along different systems” -> The context and setting of this phrase need to be clarified. The current description is overly narrative and lacks sufficient technical grounding. The entire section describing Study 3 should be restructured into a clear, step-by-step format, ensuring that each methodological stage is explicitly detailed and easy to follow.
We have clarified the section starting at line 242 to line 347. Step by step details included in the subsequent paragraphs. The introductory paragraphs serve to motivate why specific tasks were chosen.
15. Confusing that Figure 2 is present in Pg. 4 but also in Pg. 10. This duplication makes questionable whether the correct number format is mentioned in the text, such as in the section “Motor Activity (Wearable Sensors Accelerometer Data)”.
Figure numbers have been corrected
16. Figure 2 (Pg.10) -> Measurement units are missing. The font is very small and texts is unreadable. Correct this discrepancy in all other figures in methodological section (Figures 2 to 6).
We have increased the font and clarified the following: Mean Deviations (panel C), frequency (panel D), and Gamma parameters (panel E) are unitless. These are part of the standardization that we do. Panels B and C are measured across time (seconds) and IMU Accel is measured in g (9.8 m/s^2), this is illustrated at the bottom of each panel for clarity and explained in the captions and main text.
17. The signal processing steps illustrated in Figures 2 to 6 are overloaded with information and presented in a highly narrative style, making it difficult to clearly identify the key output parameters computed for each signal across Studies 1, 2, and 3. To improve clarity, the manuscript should include a dedicated summary section within the Methods—such as a “Statistical Measures” subsection—that explicitly defines all measured parameters at each processing stage and outlines the statistical techniques used to analyze them. Additionally, it is essential to clearly specify the output classes for each study and explain how these classes are standardized across the different studies to enable consistent evaluation and comparison of results.
We have rewritten the explanations and simplified as much as possible the descriptions. Studies are unified by the MMS approach described in Figure 2.1 and in detail in the first section under Data Analysis and Figure 2.2 as an example of the use of the methods.
18. Figures 7 and 8 present 3D visualizations of the mean, standard deviation, and skewness for three facial nodes labeled V1, V2, and V3. However, the Methods section does not provide sufficient detail about these nodes or clearly specify the exact measurements they represent. A more thorough description of these facial nodes and their corresponding data should be included to ensure the figures are fully interpretable.
We have written the methods starting on line 543-571 and cited recent literature explaining in more detail the use of such methods with applications to autism.
19. Result section does not present any kind of statistical comparison between groups with reporting respective p-value estimations, which can prove that the computed metrics can be effectively used for personalized biometrics.
We have provided statistical comparisons in line 670-672, Figure 9, including the captions, and line 767, figure 16 and caption. Figure 17 also offers comparisons with significance levels (* p<0.05 and ** p<0.001) and Transfer Entropy metrics. Note that we are comparing distribution parameters undergoing changes in families of distributions empirically estimated, rather than assumed theoretical means in the significant hypothesis testing paradigm. Our methods are different. Yet, we provide non-parametric comparisons.
20. The results rely heavily on graphical presentations and lack direct comparability between studies. To facilitate evaluation and comparison, a summary table should be included that reports the means or medians along with standard deviations or interquartile ranges for all computed biometric measures. Additionally, this table should present inter-group statistical comparisons to clearly highlight significant differences across the studies (according to the previous comment).
Please see the above. Each study unveils a continuous Gamma family of probability distributions and associated parameters with ranges represented in the graphs.
21. The Introduction section of the manuscript currently contains a very limited number of references (only 8), which is insufficient for the broad and well-established field of biosignal processing applied to biometrics. Furthermore, the manuscript relies heavily on the authors’ own work, with 23 out of 61 references (38%) being self-citations. The authors should expand their literature review to include a wider range of relevant studies
We have expanded the literature review in the introduction and cited all relevant papers from the literature. In terms of our methods, they are unique, and the self-citations are to establish the validation that we have done in peer reviewed journals and patents over the span of more than two decades. We have brought citations from the lab to a minimum necessary to show validity of our new framework.
Below is a sample table summarizing ranges of various parameters for one set of tasks. The figure 2.2 already captures these ranges and while we appreciate the comment on adding the ranges, it is difficult to tabulate them for all parameters of interest.
Experimental Assay |
Parameter |
Body Part |
Task |
Param Range (Physical units) |
Time window |
Standardized Fluctuations [0,1] |
Gamma NSR range |
Physical Actions to access Volitional, Involuntary and Automatic motor control |
Linear acceleration 1g=9.8m/s2 |
Pectoralis |
Rest Point Walk |
.97-1.3 g |
250ms |
0-.083 |
3.9-4.5x10-3 3.5-3.8 x10-3 2.0-3.0 x10-3 |
Linear acceleration |
Arm |
Rest Point Walk |
.8-1.3 g |
250ms |
0-.08 |
3.1-4.3x10-3 3.0-4.0 x10-3 3.1-3.3 x10-3 |
|
Linear acceleration |
Leg |
Rest Point Walk |
.4-1.5g |
250ms |
0.08 |
4.1-4.3x10-3 4.1-5.1 x10-3 4.1-4.4 x10-3 |
Round 2
Reviewer 1 Report
Comments and Suggestions for Authors
The self-citation should be revised
1. Citation Reduction in Specific Ranges
Citations 6–13:
The total number of citations in this range should be limited to no more than 2.
Citations 15–16:
This section should include no more than 1 citation.
Citations 32-33:
This section should include no more than 1 citation.
2. Citations to Be Removed Completely
The following citations are unnecessary and should be removed from the manuscript:
Citations 25 and 26:
Citations 30, 34, and 35:
Citations 56–58, 61–62, and 72–73
Author Response
I have gone over the text and reduced our references to a bear minimum so we can provide evidence in support of our unique methods and their validation in the peer reviewed literature.
Reviewer 2 Report
Comments and Suggestions for Authors
The authors did not answer adequately to most of my revision comments.
In the response to the reviewer, I would like to see a copy of the text made in accordance to each of my questions. At present, there are serious concerns that several points have been either overlooked or inadequately addressed. To ensure transparency and facilitate a thorough review, please include the exact changes made in the manuscript in direct response to each reviewer comment.
Detailed revision remarks:
- The title: The term “scalable” is used, however, its interpretation is not explained in all relevant sections: Abstract, Methods and Results.
Authors’ response: We have now addressed the term scalable in all aspects of the abstract and across the paper and better motivated their use.
Reviewer’s second comment: This answer is not correct as the term “scalable” is found only two times in the text: page 26 (Discussion) and page 29 (Conclusions). This is insufficient justification of the basis of the methods emphasized in the title.
- The answer to Comment 2 concerning the relevance of the Abstract is not sufficient. The authors answered to the first detail (objective of the study) but other details are either incorrect or were not addressed. The following information should be added to the Abstract (Methods): This sub-section does not reveal sufficient technical details for the developments in this study. The new sentences added do not reveal details for the: signal acquisition, signal processing methods, other statistical methods, Outcomes, database, type, size. Abstract (Results): Numerical results are missing. Abstract (Conclusions): Not supported by the results. Ensure that all claims are directly grounded in the numerical results provided in the Results (Abstract).
- Introduction: The comment about irrelevance of the references: A significant part of the citations (27 out of 70, or approximately 39%) are from the authors' own research group. This level of self-citation raises concerns about objectivity and the balance of the literature review. Good scientific practice encourages the inclusion of diverse and relevant sources to provide a comprehensive and unbiased overview of the field. Excessive self-citation may indicate selective referencing and risks creating an echo chamber that overemphasizes the authors’ own contributions while potentially ignoring other important work. Typically, self-citations should be limited to no more than 5%, focusing only on studies that are directly relevant and foundational to the present work.
- Follow the journal template – All subsections (and equations) must follow the defined style for numbering format. Figures and tables must appear in the first paragraph right after their first mention in the text.
Authors’ response: Adjustments made accordingly.
Reviewer’s second comment: No adjustments made on this issue.
- Figure 1: Not enough informative for the sources of information, data flow and processing steps. The figure must be clarified with sufficient text or other details, which clarify the methods and follow the flow of information in section Materials and Methods. As this is the main block diagram of the study, I miss essential information regarding the defined Study1, Study2 and Study 3 procedures and clear description of sequential signal processing steps. In my opinion, the figure must be substantially improved.
Authors’ response: The figure is now better explained clarifying that this is a meta paradigm to show schematically the steps that each participating family took in the cases where the research was remotely performed using our lab in the box concept and recycling the sensors and protocols training from one family to the next.
Reviewer’s second comment: No adjustments made on this issue. The figure is the same as before. Furthermore, note that the quality of the figure is very low and embedded text is not readable.
- Table 1: (1) The reference number of the studies (official registry) must be added;
Authors’ response: These have been added in line 152 before table 1 and line 180.
Reviewer’s second comment: No references to the studies are added in table 1 and in the text. All studies must include reference to their official registry and/or article that presents them.
- Table 1 (2) Abbreviations must be defined; (3) Measurement units must be added;
Authors’ response: Please see the table caption now with all details added.
Reviewer’s second comment: The added table footnote is NOT corresponding to any text within Table 1. All terms used in table T and explanations in the caption and footnote must be the same.
- Table 1 (4) The last column is not informative to specific “Data types” because “data” is usually connected to specific physiologic signals or specific measurements. Define the type of the recorded signals.
Authors’ response: The data type here refers to face, voice, motor and ECG with physical units now included in the caption.
Reviewer’s second comment: The caption of the table is NOT changed. I cannot find any description regarding of the recorded signals and data types.
- Figure 2 contains excessive detail and lacks clarity. While it is intended to be a methodological overview, it includes too much information related to the results, which detracts from its primary purpose. The figure should be significantly simplified to focus on key elements such as the placement of biosensors on the human body
Authors’ response: These have been clearly explained and highlighted in yellow on the heart, bicep and thigh
Reviewer’s second comment: Figure 2 was NOT changed compared to its previous version. It is still incomprehensive.
- Figure 2: and clearly labeled blocks that follow exactly to the signal processing procedures outlined in Section 2: Materials and Methods. Its goal should be to provide a high-level overview of the processing pipeline, without delving into low-level technical details—those should be reserved for separate, dedicated figures.
Authors’ response: The pipeline for signal processing has been simplified and illustrated for one case so we expand to the cohort in the results section.
Reviewer’s second comment: Figure 2 was NOT changed compared to its previous version. It still not present the required information.
- Figure 2: Final note: always check the font size and element details if readable in the pdf file.
Authors’ response: We have made sure that the pdf fonts are visible.
Reviewer’s second comment: This is NOT true! In fact, the quality of Figure 2 appears even lower than before. The embedded text is unreadable, and the lines and graphical elements are blurred and small, making the figure difficult to interpret. The figure should be redesigned from scratch, with improved resolution, appropriately scaled fonts, and clearer graphical elements to ensure readability and proper understanding by the reader.
- Ln 142 contradicts Ln 119: There is a lack of clarity regarding the ethical approvals referenced in the manuscript. It is confusing that different approvals from the same institution appear to be associated with the same study.
Authors’ response: We clarified that this is the same protocol and explicitly stated the date and type of protocol.
Reviewer’s second comment: The ethical approval information in lines 184–185 duplicates content already presented in lines 154–155. Repetition of such critical information is not acceptable, as it reflects inconsistencies in the writing of the study protocol and raises concerns about editorial oversight. The whole text in this section must be rewritten.
- The author must explicitly specify how many distinct approvals were obtained, which ethical committees granted them (number, dates), and how these approvals correspond to each sub-study within the overall research. This information is essential to ensure transparency and compliance with ethical standards.
Authors’ response: This is explained in line 180 – “We note here that this is a blanket IRB protocol to work with digital data and unconstrained motions occurring at the lab, and/or remotely at home, schools and clinics.”
Reviewer’s second comment: The statement is not comprehensive and requires further clarification. Specifically, the scope and limitations of the "blanket IRB protocol" should be clearly defined, including whether it covers all participant populations, data types, and study environments mentioned. Clear and transparent explanation is essential to confirm ethical compliance across all settings described.
- Section “Materials & Equipment” must be clarified. The position and type of biosensors is unknown. The type and version of the hardware/software equipment is also unknown. This section must be totally rewritten.
Authors’ response: Section clarified with system details and sensor types/positions.
Reviewer’s second comment: The changes made are unsatisfactory and inappropriate. The authors decided to change the caption of Figure 2.1 instead of adding explanations in the main text. As a result the caption is excessively long and includes methodological details that should be part of the main text. The main text is still very short and not informative. This section must be totally rewritten.
- Ln 202: “biorhythms along different systems” -> The context and setting of this phrase need to be clarified. The current description is overly narrative and lacks sufficient technical grounding. The entire section describing Study 3 should be restructured into a clear, step-by-step format, ensuring that each methodological stage is explicitly detailed and easy to follow.
Authors’ response: We have clariied the section starting at line 242 to line 347. Step by step details included in the subsequent paragraphs. The introductory paragraphs serve to motivate why specific tasks were chosen.
Reviewer’s second comment: I cannot see changes in these lines related to this question, which must clarify “biorhythms along different systems”. The overall explanation is still overly narrative and lacks sufficient technical grounding.
- Confusing that Figure 2 is present in Pg. 4 but also in Pg. 10. This duplication makes questionable whether the correct number format is mentioned in the text, such as in the section “Motor Activity (Wearable Sensors Accelerometer Data)”.
Authors’ response: Figure numbers have been corrected.
Reviewer’s second comment: Figures numbers were not corrected properly. The current numbering (e.g., Figure 2.1, Figure 2.2) does not comply with the journal’s formatting guidelines, which require the use of simple integer-based figure numbering (e.g., Figure 2, Figure 3, etc.).
- Figure 2 (Pg.10) -> Measurement units are missing. The font is very small and texts is unreadable. Correct this discrepancy in all other figures in methodological section (Figures 2 to 6).
Authors’ response: We have increased the font and clarified the following: Mean Deviations (panel C), frequency (panel D), and Gamma parameters (panel E) are unitless. These are part of the standardization that we do. Panels B and C are measured across time (seconds) and IMU Accel is measured in g (9.8 m/s^2), this is illustrated at the bottom of each panel for clarity and explained in the captions and main text..
Reviewer’s second comment: The resolution of ALL figures in the text very low and not appropriate for publication. Furthermore, I don’t see any changes made according to my revision comments. The figures are the same as the first article submission version.
- The signal processing steps illustrated in Figures 2 to 6 are overloaded with information and presented in a highly narrative style, making it difficult to clearly identify the key output parameters computed for each signal across Studies 1, 2, and 3. To improve clarity, the manuscript should include a dedicated summary section within the Methods—such as a “Statistical Measures” subsection—that explicitly defines all measured parameters at each processing stage and outlines the statistical techniques used to analyze them. Additionally, it is essential to clearly specify the output classes for each study and explain how these classes are standardized across the different studies to enable consistent evaluation and comparison of results.
Authors’ response: We have rewritten the explanations and simpliied as much as possible the descriptions. Studies are uniied by the MMS approach described in Figure 2.1 and in detail in the first section under Data Analysis and Figure 2.2 as an example of the use of the methods.
Reviewer’s second comment: I don’t see any relevant changes made in respect to this revision comment. The authors should provide a list of all changes made in respect to this question.
- Result section does not present any kind of statistical comparison between groups with reporting respective p-value estimations, which can prove that the computed metrics can be effectively used for personalized biometrics.
Authors’ response: We have provided statistical comparisons in line 670-672, Figure 9, including the captions, and line 767, figure 16 and caption. Figure 17 also offers comparisons with significance levels (* p<0.05 and ** p<0.001) and Transfer Entropy metrics. Note that we are comparing distribution parameters undergoing changes in families of distributions empirically estimated, rather than assumed theoretical means in the significant hypothesis testing paradigm. Our methods are different. Yet, we provide non-parametric comparisons.
Reviewer’s second comment: All statistical methods, including any non-parametric comparisons, must be clearly described in the Methods section. Additionally, any claims of statistical significance in the Results should be explicitly linked to the corresponding methods, including the specific tests used. It is also essential to report the sample sizes or data subsets on which these tests were performed to allow proper interpretation of the results and their validity. Without this information, the statistical relevance and robustness of the findings remain unclear.
- The results rely heavily on graphical presentations and lack direct comparability between studies. To facilitate evaluation and comparison, a summary table should be included that reports the means or medians along with standard deviations or interquartile ranges for all computed biometric measures. Additionally, this table should present inter-group statistical comparisons to clearly highlight significant differences across the studies (according to the previous comment).
Authors’ response: Please see the above. Each study unveils a continuous Gamma family of probability distributions and associated parameters with ranges represented in the graphs.
Reviewer’s second comment: No action has been done by the authors to answer this comment. It is true that this study lack essential numerical results summarized in a table with valid intergroup comparisons.
Additional comment: The starting paragraph and Table 1 in section “2. Materials and Methods” should be encapsulated within a clearly labeled subsection. Proper structuring enhances clarity and allows readers to easily navigate the content.
Author Response
Detailed revision remarks:
- The title: The term “scalable” is used, however, its interpretation is not explained in all relevant sections: Abstract, Methods and Results.
Authors’ response: We have now addressed the term scalable in all aspects of the abstract and across the paper and better motivated their use.
Reviewer’s second comment: This answer is not correct as the term “scalable” is found only two times in the text: page 26 (Discussion) and page 29 (Conclusions). This is insufficient justification of the basis of the methods emphasized in the title.
Author’s second comment
The term scalable is explained in the abstract on line 11-13 and highlighted in yellow. I quote “There is an emerging need for new scalable behavioral assays, i.e., assays that are feasible to administer from the comfort of the person’s home, with ease and at higher frequency than clinical visits or visits to laboratory settings can afford us today.”
It is explained once more from line 13-14. I quote and highlight in yellow “This need poses several challenges which we address in this work along with scalable solutions for behavioral data acquisition and analyses aimed at diversifying various populations under study here and to encourage citizen-driven participatory models of research and clinical practices.”
On line 80 here we offer technologies that enable remote assessment on a scale and with high frequency, from the comfort of one’s home.
On line 84 In turn, this scaling of our scientific quests to include underserved sectors of the population has the potential to dampen the various biases that we currently face when developing new technologies and treatments in trackers of digital health and wellness.
On line 91 Co-registering data and scaling it according to individualized anatomical features under our micro-movement spikes approach 13, offers new ways to assess behaviors across a multiplicity of sampling resolutions and sensors signals types.
The Methods subsection of the abstract further describes scalability “These methods can be applied to various research programs ranging from clinical trials at home, to remote pedagogical settings. They are aimed at creating new standardized biometric scales to screen and diagnose neurological disorders across the human lifespan.”
On line 152 Ultimately, such work helps us evaluate the ecological validity of our current approaches – as participants are acting and behaving within more naturalistic settings – and allows for large scale, truly inclusive studies via the use of commercially available wearable sensing technologies that most people carry around.
Etc.
To highlight the sense of scalability we do not need to repeat the word scalable or scalability throughout the paper. It is precisely described in the work rather than repeated frequently across the text. The entire introduction defines what scalable means.
Scalable means that we can sample the population more broadly using a combination of brief, simple-to-deploy assays and off-the-shelf, affordable tools, such as smart phones, smart watches and other commercially available tools that most people carry around. It is highly nontrivial to achieve true scalability that is autonomously deployable. That is what this paper is about. It is explained all through the paper and methods, discussion, etc. and backed up by our peer-reviewed work which we must cite, due to its uniqueness. There is a non-obvious element in the data processing and analysis that distinguishes this work from the rest of the field. We have explained it in the figures and methods, and we have cited the relevant references. Our lab has produced over 130 papers on the matter, 9 patents, 4peer reviewed books and 3 e-books in Frontiers, one e-book in this very journal. We have only cited a subset of that work to highlight the validation of the methods by peer-reviewed and blind reproducibility of our new techniques. Most of the cited papers are collaborative works with both domestic and international labs ranging from 2-7 labs. When we cite our work, we cite the work of many people that collaborate in this endeavor.
- The answer to Comment 2 concerning the relevance of the Abstract is not sufficient. The authors answered to the first detail (objective of the study) but other details are either incorrect or were not addressed. The following information should be added to the Abstract (Methods): This sub-section does not reveal sufficient technical details for the developments in this study. The new sentences added do not reveal details for the: signal acquisition, signal processing methods, other statistical methods, Outcomes, database, type, size.
Author’s second comment
We include in the Methods section of the Abstract the number of participants, age range and demographic distribution of males and females). We also include the types of time series biorhythmic data that we present and mention the unique methods designed by our laboratory, which we will describe in detail in the Methods section of the paper. Beyond that level of detail, we will exceed the word limits of the journal. This is a different paper than most regular papers in that it describes a general methodology to scale scientific practices and provides 3 examples of how to do so in the classroom, laboratory and home environment. We have many more studies that we could present as examples. However, these should be sufficient to illustrate the methods. “Methods: Our methods are centered on the biophysical fluctuations unique to the person and on the characterization of behavioral states using standardized biorhythmic time series data (from kinematic, electrocardiographic, voice, and video-based tools) in naturalistic settings, outside a laboratory environment. The methods are illustrated with 3 representative studies (60 participants, 8 to 70 years old, 34 males, 26 females). Data is presented across the nervous systems under a proposed functional taxonomy that permits data organization according to nervous systems’ maturation and decline levels. These methods can be applied to various research programs ranging from clinical trials at home, to remote pedagogical settings. They are aimed at creating new standardized biometric scales to screen and diagnose neurological disorders across the human lifespan.”
Abstract (Results): Numerical results are missing.
Numerical results and explanation are added on lines 31-40 and highlighted in yellow “Each study provides parameter spaces with self-emerging clusters whereby data points corresponding to a cluster are probability distribution parameters automatically classifying participants into different continuous Gamma probability distribution families. Non-parametric analysis reveals significant differences in distributions’ shape and scale (p<0.01). Data reduction is realizable from full probability distribution families to a single parameter, the Gamma scale, amenable to represent each participant within each subclass, and each cluster of similar participants within each cohort. We report on data integration from stochastic analyses that serve to differentiate participants and propose new ways to highly scale our research, education, and clinical practices.”
Abstract (Conclusions): Not supported by the results. Ensure that all claims are directly grounded in the numerical results provided in the Results (Abstract).
Author’s second comment
The Conclusions are supported by the results from these three studies and by a body of research (not included here) that our lab has created over the span of more than 2 decades.
We cannot include any more details as this is a methods paper with 3 representative studies.
- Introduction: The comment about irrelevance of the references: A significant part of the citations (27 out of 70, or approximately 39%) are from the authors' own research group. This level of self-citation raises concerns about objectivity and the balance of the literature review.
Author’s second comment
Another way to look at is that we have cited 27 papers our of 130 peer reviewed papers that the lab has published and more than 20 peer reviewed chapters across multiple books on the matter, 9 patents licensed to companies and other labs’ blind reproduction of our work. That represents less than 16% of unique work across multiple fields. More importantly, we have cited the relevant extant literature to situate our work in the context of what has been done in the various areas of work. This is not a review paper but rather a methods paper presenting examples of how to scale scientific experiments using new methods that we have invented and validated in 1000s of people across the human lifespan. We only had room for a selective representative set of studies to illustrate the methods.
- Good scientific practice encourages the inclusion of diverse and relevant sources to provide a comprehensive and unbiased overview of the field. Excessive self-citation may indicate selective referencing and risks creating an echo chamber that overemphasizes the authors’ own contributions while potentially ignoring other important work. Typically, self-citations should be limited to no more than 5%, focusing only on studies that are directly relevant and foundational to the present work.
Author’s second comment
Indeed, when the work is innovative and disruptive, such as our own work, it is important to highlight how it differs from the extant literature. To that end, we have cited the prior work and explained how our current techniques differ in fundamental ways from current approaches. We cannot do 5% self-citations because the work is more extensive than one single study and spans multiple fields. Furthermore, the work is collaborative, and it involves multiple labs. We are not interested in inflating the h-index. We are interested in providing the means to verify that what we are presenting has been validated by the peer review system.
We have covered in our 3 examples different contextual situations, different populations (ages) and different pathologies of the nervous systems. It is therefore imperative to back up our results with peer-reviewed literature that includes our own work and the work of others who have reproduced out methods. Those are in addition to the prior literature which does their analyses differently, using a one-size-fits-all approach to highly heterogeneous conditions.
- Follow the journal template – All subsections (and equations) must follow the defined style for numbering format. Figures and tables must appear in the first paragraph right after their first mention in the text.
Authors’ response: Adjustments made accordingly.
Reviewer’s second comment: No adjustments made on this issue.
Author’s second comment
Each figure and table is cited as close as possible to the mentioning in the text. During typesetting, the journal’s production office will move the figures and tables around to accommodate the text. The word template is not the final form of the paper. This is a draft of what it will look like (approximately).
All subsections have been numbered and the styles template of the MDPI journal used as provided.
- Figure 1: Not enough informative for the sources of information, data flow and processing steps. The figure must be clarified with sufficient text or other details, which clarify the methods and follow the flow of information in section Materials and Methods. As this is the main block diagram of the study, I miss essential information regarding the defined Study1, Study2 and Study 3 procedures and clear description of sequential signal processing steps. In my opinion, the figure must be substantially improved.
Authors’ response: The figure is now better explained clarifying that this is a meta paradigm to show schematically the steps that each participating family took in the cases where the research was remotely performed using our lab in the box concept and recycling the sensors and protocols training from one family to the next.
Reviewer’s second comment: No adjustments made on this issue. The figure is the same as before. Furthermore, note that the quality of the figure is very low and embedded text is not readable.
Author’s second comment
The figure is the same as before, but has been modified to enlarge the font. The explanation in the caption was changed to include more details. It is important to maintain this figure as it is because it is a high-level schematic of a general formulation that we are proposing and that we have successfully implemented already in longitudinal studies involving both clinical trials and basic research.
Figure 1 is mentioned on line 103 and presented in line 105, close to the referencing paragraph.
- Table 1: (1) The reference number of the studies (official registry) must be added;
Authors’ response: These have been added in line 152 before table 1 and line 180.
Reviewer’s second comment: No references to the studies are added in table 1 and in the text. All studies must include reference to their official registry and/or article that presents them.
Author’s second comment
The reference number of the study was added. I quote and highlight in yellow “All three studies (Table 1) took place at the Sensory Motor Integration Laboratory of Rutgers University and/or remotely at the home of the participant. The study protocol was approved by the Rutgers University Institutional Review Board (IRB) [Study ID: Pro2020000154] on 6/09/2023. ”
This is stated on line 159 and Table 1 starts on line 164. All three studies are under the same IRB protocol. The figures and analyses have not been published so there is no associated reference (paper to the data here).
As we explained before, this is a blanket IRB protocol to enable remote testing using wearables and video-based technologies. The protocol is under US HIPAA act compliance and approved by the university legal team, such that data go to our cloud and protects participants privacy. Furthermore, the participants have a say on data ownership and the model is under the current philosophy of participatory research. This is further mention in line 193.
- Table 1 (2) Abbreviations must be defined; (3) Measurement units must be added;
Authors’ response: Please see the table caption now with all details added.
Reviewer’s second comment: The added table footnote is NOT corresponding to any text within Table 1. All terms used in table T and explanations in the caption and footnote must be the same.
Author’s second comment
The Table 1 had words with symbols (+, ++, +++) corresponding to the type of data and type of instrumentation with physical units described in the caption. To add clarity, we have now placed in parenthesis the data type in the last column of the table.
- Table 1 (4) The last column is not informative to specific “Data types” because “data” is usually connected to specific physiologic signals or specific measurements. Define the type of the recorded signals.
Authors’ response: The data type here refers to face, voice, motor and ECG with physical units now included in the caption.
Reviewer’s second comment: The caption of the table is NOT changed. I cannot find any description regarding of the recorded signals and data types.
Author’s second comment
+Video (face) data in f/s, ++Microphone (Voice) data unitless, +++Accelerometer inertial measurement units IMUs (Motor) data in m/s2, ++++(electro-cardiography ECG (Hear) in mV/s), FUME stands for Facial Universal Micro Expressions, PD stands for Parkinson’s Disease, TD stands for Typically Developing, ASD stands for Autism Spectrum Disorders, ET stands for Essential Tremor
- Figure 2 contains excessive detail and lacks clarity. While it is intended to be a methodological overview, it includes too much information related to the results, which detracts from its primary purpose. The figure should be significantly simplified to focus on key elements such as the placement of biosensors on the human body
Authors’ response: These have been clearly explained and highlighted in yellow on the heart, bicep and thigh
Reviewer’s second comment: Figure 2 was NOT changed compared to its previous version. It is still incomprehensive.
Author’s second comment
The Figure 2 and the comments regarding it were meant to remain as they are. It is very important to maintain this figure as is. This figure reflects the integration of multiple data streams under the same personalized statistical platform. The style used in this figure is common style of Nature, Science, Neuron, PNAS and other high impact journals that explain analytical pipelines and where we have previously published such type of work. The fonts and scales can be zoomed in when the PDF version is provided. This style of figure is common practice in STEM fields.
- Figure 2: and clearly labeled blocks that follow exactly to the signal processing procedures outlined in Section 2: Materials and Methods. Its goal should be to provide a high-level overview of the processing pipeline, without delving into low-level technical details—those should be reserved for separate, dedicated figures.
Authors’ response: The pipeline for signal processing has been simplified and illustrated for one case so we expand to the cohort in the results section.
Reviewer’s second comment: Figure 2 was NOT changed compared to its previous version. It still not present the required information.
Author’s second comment
The pipeline specificity and simplified is in Figure 3 now (formerly Figure 2.2, which we named after zooming in Figure 2.1 for the specific IMU data (acceleration in g units or 9.8m/s2)
- Figure 2: Final note: always check the font size and element details if readable in the pdf file.
Authors’ response: We have made sure that the pdf fonts are visible.
Reviewer’s second comment: This is NOT true! In fact, the quality of Figure 2 appears even lower than before. The embedded text is unreadable, and the lines and graphical elements are blurred and small, making the figure difficult to interpret. The figure should be redesigned from scratch, with improved resolution, appropriately scaled fonts, and clearer graphical elements to ensure readability and proper understanding by the reader.
Author’s second comment
We do not understand this comment.
The pdf version of the paper that we get is readable and the fonts are visible particularly when we zoom in. All axes are labeled correctly, and all units are expressed on the corresponding axis.
We have re-uploaded the figures in higher resolution for production but perhaps the pdf conversion degrades them in this step of the review process.
- Ln 142 contradicts Ln 119: There is a lack of clarity regarding the ethical approvals referenced in the manuscript. It is confusing that different approvals from the same institution appear to be associated with the same study.
Authors’ response: We clarified that this is the same protocol and explicitly stated the date and type of protocol.
Reviewer’s second comment: The ethical approval information in lines 184–185 duplicates content already presented in lines 154–155. Repetition of such critical information is not acceptable, as it reflects inconsistencies in the writing of the study protocol and raises concerns about editorial oversight. The whole text in this section must be rewritten.
Author’s second comment
We do not understand this comment.
The Rutgers Institutional Review (IRB) protocol in compliance with the Helsinki act and with the regulatory bodies ensuring HIPAA laws compliance are reflected in line 159 All three studies (Table 1) took place at the Sensory Motor Integration Laboratory of Rutgers University and/or remotely at the home of the participant. The study protocol was approved by the Rutgers University Institutional Review Board (IRB) [Study ID: Pro2020000154] on 6/09/2023.
- The author must explicitly specify how many distinct approvals were obtained, which ethical committees granted them (number, dates), and how these approvals correspond to each sub-study within the overall research. This information is essential to ensure transparency and compliance with ethical standards.
Authors’ response: This is explained in line 193 – “We note here that this is a blanket IRB protocol to work with digital data and unconstrained motions occurring at the lab, and/or remotely at home, schools and clinics.”
Reviewer’s second comment: The statement is not comprehensive and requires further clarification. Specifically, the scope and limitations of the "blanket IRB protocol" should be clearly defined, including whether it covers all participant populations, data types, and study environments mentioned. Clear and transparent explanation is essential to confirm ethical compliance across all settings described.
Author’s second comment
All three studies (Table 1) took place at the Sensory Motor Integration Laboratory of Rutgers University and/or remotely at the home of the participant. The study protocol was approved by the Rutgers University Institutional Review Board (IRB) [Study ID: Pro2020000154] on 6/09/2023. This protocol covers all participant types, location of research and sensors.
On line 192 All participants provided informed consent for data collection and video recording approved by the same Rutgers University Institutional Review Board (IRB Study ID: Pro2020000154 approved on 6/09/2023). We note here that this is a blanket IRB protocol to work with digital data and unconstrained motions occurring at the lab, and/or remotely at home, schools and clinics. The protocol includes HIPAA compliance to protect the participants’ privacy (video and biometric data) as well as the participant’s ownership of the video and biosensors’ data.
- Section “Materials & Equipment” must be clarified. The position and type of biosensors is unknown. The type and version of the hardware/software equipment is also unknown. This section must be totally rewritten.
Authors’ response: Section clarified with system details and sensor types/positions.
Reviewer’s second comment: The changes made are unsatisfactory and inappropriate. The authors decided to change the caption of Figure 2.1 instead of adding explanations in the main text. As a result the caption is excessively long and includes methodological details that should be part of the main text. The main text is still very short and not informative. This section must be totally rewritten.
Author’s second comment
The Section contains all the rationale and motivation to place sensor locations and functional assessment through specific assays. This is explained in such detail that the reviewer states that there is too much verbiage and too many details in the main text (later on).
So, at this point we are confused as to what the reviewer wants. This paper is not written for a single reviewer. It is written for a general audience, such that the audience understands why we are doing what we are doing and knows how to reproduce it.
Usually, a review process involves constructive critique of the work so we can improve it. This is not a constructive review of our work. This is a very capricious demand on how we are supposed to present the work, even the protocol number of our study. The reviewer employs accusatory language borderline disrespecting the effort of our team and using all CAPS and even exclamation point throughout the review. We have addressed all suggestions and explicitly cited the location of the paper where the changes have been provided.
On lines 200-210, we explain the MC10 Biostamp sensor location which is also shown in Figure 2
The MC10 BioStamp-nPoint system (Lexington, MA, 2008) wireless sensors were utilized to capture biophysical data (cardiac and motor). Figure 2 shows the sensor locations on the body along with the corresponding sample waveform data outputs. Electrical activity was captured via 4 separate electrodes on each wearable sensor at a sampling frequency of 250 Hz. This included EMG measurements from the bicep of the dominant arm and ECG measurements from the sensor placed on the lead II position of the chest. Triaxial acceleration was recorded for all three sensors locations – chest lead II position, bicep of the dominant arm, and the leg (upper thigh) opposite the dominant arm (for example, on the left leg if the participant is right-handed) at a sampling rate of 31.25Hz. A charging station along with a Samsung phone were utilized to calibrate the sensors and timestamp the experimental tasks.
On line 211-217 we explain the video-based capture
A webcam (Logitech - C920) was utilized to capture video data (30Hz), including facial data, which was later analyzed via the OpenPose and OpenFace programs from Carnegie Mellon groups 26 to assess motor activity and facial expression. A mini tripod for the webcam was also provided. An isolated microphone was utilized to capture voice data. A standard laptop with internet and microphone capabilities was provided if needed. An expandable camera stand/tripod was also provided for the Pointing task of the third study.
This section provides references to the relevant software and companies/academic groups such that people can reproduce what is presented here.
- Ln 202: “biorhythms along different systems” -> The context and setting of this phrase need to be clarified. The current description is overly narrative and lacks sufficient technical grounding. The entire section describing Study 3 should be restructured into a clear, step-by-step format, ensuring that each methodological stage is explicitly detailed and easy to follow.
Authors’ response: We have clarified the section starting at line 242 to line 347. Step by step details included in the subsequent paragraphs. The introductory paragraphs serve to motivate why specific tasks were chosen.
Reviewer’s second comment: I cannot see changes in these lines related to this question, which must clarify “biorhythms along different systems”. The overall explanation is still overly narrative and lacks sufficient technical grounding.
Author’s second comment
Biorhythms are a common word in the field. They are capturable waveforms using a variety of sensors and video-based means (through computer vision methods) involving time series of values, often in the form of peaks and valleys with different frequencies, periods and time scales. We explain this in the context of a functional taxonomy of the nervous system that our lab has created for the purpose of integrating different levels of autonomy and control. We have applied these principles to autism and Parkinson’s disease and detailed them in various papers and books which we do not cite here. The technical foundation of this body of work comes from our lab but we refrained from citing it. We have now cited a few papers (a small subset) to justify the proposed paradigm which is used in the ambulatory context of homes and clinics (outside the constraints of a lab.)
Here is the edited text with the technical references included
The third study involved 18 participants (9 males and 9 females), mean age 20.2 with standard deviation of 10.2 years of age. Of the 18 participants, 10 were typically developing (TD) and 8 were autistics. Of the 8 ASD participants, 2 were diagnosed with a SYNGAP1 mutation often associated with autistic traits and delays in neurodevelopment 1. To assess different areas of functionality, autonomy and control, our lab has designed a series of tasks that while brief (ranging between 5seconds to 3minutes) permit the digital evaluation of the participant using commercially available means, from the comfort of their home. Every task described below was first assessed with high grade sensors in the laboratory to determine optimal time windows to use with lower grade sensors that are commercially available. The combination of brief functional assays and signals from off-the-shelf instruments make our methods highly scalable.
We recorded biorhythms registered by sensors in Figures 2-3 and Table 1 along different systems (motor & facial – peripheral, and cardiac – autonomic) and used a compact set of brief experimental assays designed and previously tested in our lab 2,3 to evoke signals relevant to the assessment of the systems involved at different neurophysiological levels. These spanned from functions that are vital to the person’s survival (such as heart rate variability), to functions that are important for activities of daily living and the overall person’s well-being. The latter included assessment of involuntary motions that are invisible to the naked eye 4. An example is undesirable involuntary motions at rest while the person is trying to remain still upon being instructed to not move. We can isolate the physiologically relevant signals and measure the volitional control of the brain over the body at rest 5,6. Then we can assess the level of involuntary jitter, tics, and other motions bound to interfere with overall motor control. We will refer to this as the resting task.
An important task that enables us to also measure cognitive-motor performance is one that involves the voluntary control of hand motions during pointing and reaching behaviors. The hands can broadcast our wishes and / or even our hesitations to communicate something to another person in a controlled manner. They can also provide insight into the connection between mental intent and physical realization of that intent. This allows us to observe how much of the person's action is driven by volition, in a deliberate manner, vs. how much is spontaneously supported, automatically—without adding cognitive load 5,7. In this regard, we measure within pointing movements to a visual target, not only the forward segment intended to touch the target, but more importantly, we measure the quality of the backward segment – occurring spontaneously (without instruction). This segment of the reach, which is hidden to the naked eye of an observer and often occurring largely beneath the awareness of the performer 7,8, brings the hand to rest and connects the sequencing of the motion to the target with the pauses in route to the next goal-directed segment. We refer to this later as the pointing task.
We can also measure the amount of automatic control that a person has over their movements – particularly movements that enable the person to be independent and ambulatory, free of the need to be assisted by another person. For example, measuring how an individual walks independently and takes turns that require balancing the body up against gravity (as well as continuing to have a fluid gait pattern after turning) can inform us about the person’s well-being in daily living. Extensive work in our lab has derived multiple personalized biometrics of gait. A subset of that work, which we reproduce here in different settings is provided 9-11. We later refer to this as the walking task.
- Confusing that Figure 2 is present in Pg. 4 but also in Pg. 10. This duplication makes questionable whether the correct number format is mentioned in the text, such as in the section “Motor Activity (Wearable Sensors Accelerometer Data)”.
Authors’ response: Figure numbers have been corrected.
Reviewer’s second comment: Figures numbers were not corrected properly. The current numbering (e.g., Figure 2.1, Figure 2.2) does not comply with the journal’s formatting guidelines, which require the use of simple integer-based figure numbering (e.g., Figure 2, Figure 3, etc.).
Author’s second comment
Figure numbers have been corrected to Figure 1, 2, 3, etc. and cited in the text according to the order
- Figure 2 (Pg.10) -> Measurement units are missing. The font is very small and texts is unreadable. Correct this discrepancy in all other figures in methodological section (Figures 2 to 6).
Authors’ response: We have increased the font and clarified the following: Mean Deviations (panel C), frequency (panel D), and Gamma parameters (panel E) are unitless. These are part of the standardization that we do. Panels B and C are measured across time (seconds) and IMU Accel is measured in g (9.8 m/s^2), this is illustrated at the bottom of each panel for clarity and explained in the captions and main text.
Reviewer’s second comment: The resolution of ALL figures in the text very low and not appropriate for publication. Furthermore, I don’t see any changes made according to my revision comments. The figures are the same as the first article submission version.
Author’s second comment
Figure 2 has units in each panel where physical units are appropriate, i.e., along the raw data (acceleration (g), emg mV, ECG mV) and along the horizontal axis frame #. The Micro-movement spikes (MMS) are standardized and fall in the real range of [0,1]. The last panel with Analytics presents 2 parameter spaces. These parameters are unitless and standardized. We have recreated the figures in a and re-uploaded them. The pdf process may degrade them but the production process is different as it includes the HR figures.
- The signal processing steps illustrated in Figures 2 to 6 are overloaded with information and presented in a highly narrative style, making it difficult to clearly identify the key output parameters computed for each signal across Studies 1, 2, and 3. To improve clarity, the manuscript should include a dedicated summary section within the Methods—such as a “Statistical Measures” subsection—that explicitly defines all measured parameters at each processing stage and outlines the statistical techniques used to analyze them. Additionally, it is essential to clearly specify the output classes for each study and explain how these classes are standardized across the different studies to enable consistent evaluation and comparison of results.
Authors’ response: We have rewritten the explanations and simplified as much as possible the descriptions. Studies are unified by the MMS approach described in Figure 2.1 and in detail in the first section under Data Analysis and Figure 2.2 as an example of the use of the methods.
Reviewer’s second comment: I don’t see any relevant changes made in respect to this revision comment. The authors should provide a list of all changes made in respect to this question.
Author’s second comment
The Section Data Analysis describes in detail the steps of the pipeline with reference to Figure 3 which zooms in the accelerometry across body parts. The first panel A provides the sensors locations. The second column in panel B provides the raw data, which is then used to compute the deviations from the empirically estimated mean and the inset shows the standardization (personalized to account for anatomical disparities) such that multiple subjects of different ages can be placed on the same parameter space and examined across different pathologies. This is the personalized approach that our lab invented for personalized medicine. Panel D provides the histograms of the MMS which are derived from the standardization of the mean deviations. Example of MMS are in Figure 2 under MMS Standardization. Those red peaks of the MMS are gathered in the frequency histogram and the parameters of the continuous Gamma family empirically estimated to provide the parameter spaces of interest across different participants, across different ages and bodily sizes.
We explain all of it under the Data Analysis section, which references the Figure 3. In this section we cite our work heavily because there is no other work like it. We invented these methods and have tested them in thousands of participants, from neonates to the elderly and across autism, cerebral palsy, ADHD, various rate genetic disorders (Fragile X, SHANK3 deletion, SYNGAP1, etc.) schizophrenia and human transcriptome analysis (beyond the scope of this paper but relevant to the Precision Medicine paradigm which we have non-trivially modified to integrate disparate data types).
The derivation of the MMS is explained from 393 to 410. Then the estimation of probability distribution families through MLE and the determination of the continuous Gamma family of distributions starts from line 411 to line 438. These methods have been established and patented by the US, Canada, the European Union (Germany and France) and the UK. They are in use and have been reproduced across labs in Israel and the US.
We also note that several of the “self-citations” are work in collaboration with other labs, as our work spans multiple areas of research worldwide and US wide, so when we cite the work, it is not exactly self-citation but rather an example of large collaborations. The patented work is from the senior author, however, those published patents have numerous citations of prior art as we build a strong case for innovation: novelty (new science), non-obviousness (science that escaped people and make the invention unique beyond novelty) and physical embodiment (implementation in a physical device beyond algorithms and mathematical formulae). The work is already deployable at scale because of these elements. People can use an app that instantaneously provides feedback on dysregulation or regulation states of the face and heart (for example). But these are beyond the scope of this paper. Here we are merely describing the lab in the box concept.
Torres, E. B. System and method for determining amount of volition in a subject. US patent (2017, Oct. 19).
European Patent Office https://patents.google.com/patent/EP3528699A4/en
US Patent Office https://patents.google.com/patent/US10786192B2/fr
Torres, E. B. System and Method for measuring physiologically relevant motion. US patent (2017).
European Patent office https://patents.google.com/patent/EP3229684A4/en
US Patent office https://patents.google.com/patent/US20170340261A1/en
- Result section does not present any kind of statistical comparison between groups with reporting respective p-value estimations, which can prove that the computed metrics can be effectively used for personalized biometrics.
Authors’ response: We have provided statistical comparisons in line 725-727, 730-733, Figure 10, including the captions, and line 825-831, figure 17 and caption. Figure 18 also offers comparisons with significance levels (* p<0.05 and ** p<0.001) and Transfer Entropy metrics. Note that we are comparing distribution parameters undergoing changes in families of distributions empirically estimated, rather than assumed theoretical means in the significant hypothesis testing paradigm. Our methods are different from traditional methods. Yet, we provide non-parametric comparisons.
Reviewer’s second comment: All statistical methods, including any non-parametric comparisons, must be clearly described in the Methods section. Additionally, any claims of statistical significance in the Results should be explicitly linked to the corresponding methods, including the specific tests used. It is also essential to report the sample sizes or data subsets on which these tests were performed to allow proper interpretation of the results and their validity. Without this information, the statistical relevance and robustness of the findings remain unclear.
Author’s second comment
The parameters of interest in these methods are probability distribution parameter ranges. Each point on the stochastic parameter spaces of interest represents an entire probability distribution of MMS values derived from raw data. Each distribution comes from large number of measurements (>100 peaks) which are based on the sampling resolution of the sensors in use. For example, for 250Hz, 10 minutes of data provide 250x60x10 values. For video based at 30Hz 5 seconds offer 150 points per each 5second-long window, both of which offer sufficient power for statistical analysis. We are not following the significant hypothesis testing paradigm on biophysical raw data points as the points in our model are distributions themselves and according to our studies, they follow law-like relations leading to data reduction. In fact, instead, we compute and visualize confidence intervals of the parameter estimates (e.g., Figure 11,15) to assess the significance of the parameter, which is mentioned in the methods section line 459.
To further expand on this point, we have now provided Appendix Figures 1 and 2 with reference to the face data analysis. We use an appropriate distance metric in a probability space endowed with this metric to show the differences between families of probability distribution functions in the continuous Gamma family that we have obtained empirically through MLE and at 95% confidence for each shape and scale parameter of the Gamma PDF. In Appendix Figure 1 we show the representation of the distances between each point along the stochastic trajectory in Figure 8 (red V1, blue V2 and black V3) and the reference emotional signatures derived from Vanessa Van Edwards as the reference subject. This graph highlights which reference emotions are closest to the trajectory of the participant and which are farthest. Since this is done for each point along the stochastic trajectory (i.e., for each Gamma PDF) and since each point in this trajectory contains 150 measurements, we have enough statistical power and tight 95% confidence regions in the estimation of the gamma parameters (shape and scale) giving us the Gamma moments (mean, var, skewness and kurtosis) used in this parameter space of Figure 8. Then Appendix Figure 2 shows the histograms of each emotion comparison and the matrices of pairwise non-parametric stats test (rank sum) to obtain a p-value and determine the significance of each distance measurement, i.e., how far apart each region in the stochastic trajectory is from the reference FUME signatures.
Importantly, we have added Subsection 2.3.5 Statistical Tests starting on line 650 explaining the statistical tests and distance metric for probability spaces
- The results rely heavily on graphical presentations and lack direct comparability between studies. To facilitate evaluation and comparison, a summary table should be included that reports the means or medians along with standard deviations or interquartile ranges for all computed biometric measures. Additionally, this table should present inter-group statistical comparisons to clearly highlight significant differences across the studies (according to the previous comment).
Authors’ response: Please see the above. Each study unveils a continuous Gamma family of probability distributions and associated parameters with ranges represented in the graphs.
Reviewer’s second comment: No action has been done by the authors to answer this comment. It is true that this study lack essential numerical results summarized in a table with valid intergroup comparisons.
Author’s second comment
See Figure 10 panels B, C and D where parameter ranges are delineated and expressed as box-whisker graphs with p values reported from comparisons between neurotypical (NT) and patients with Parkinson’s Disease (PD). Values are also reported in the caption and text. It is impossible to tabulate hundreds of values of distribution parameters estimated from thousands of data points from one person and from a cohort. These methods are different from the traditional statistics that the reviewer refers to. Each representative study is an example of how to use our methods.
Figure 10 B-C-D depicts statistical comparisons and reports p-values for the voice analysis
Figure 17 summarizes the statistical comparisons of the Gamma moments and Gamma parameters that were empirically estimated across subjects for the standardized micro movement spikes derived from the ECG data. Furthermore, p-values are reported in the caption and in the main text (lines 835).
Figure 18 also reports the comparisons with violin plots and highlights the significance of the statistical differences in the case of Transfer Entropy values, p-values reported in the caption and main text line863.
Appendix Figure 2B reports on p-values derived from the EMD of PDFs, main text line 733
Additional comment: The starting paragraph and Table 1 in section “2. Materials and Methods” should be encapsulated within a clearly labeled subsection. Proper structuring enhances clarity and allows readers to easily navigate the content.
Author’s second comment
All sections and subsections have been numbered.
Round 3
Reviewer 2 Report
Comments and Suggestions for Authors
I believe that the two rounds of revision have helped the authors improve their manuscript, despite their evident confidence and high self-assessment regarding their experience and the significance of their work. Ultimately, it is important that the text is presented with appropriate clarity and simplicity for the readers, and that figures and tables are sufficiently clear and informative.
I understand that this article combines developments from three separate studies, which makes it quite broad and, at times, difficult to follow at first glance. It would be beneficial if the authors, especially the corresponding author, approach the reviewers' comments with more openness and understanding. A reviewer serves as an external observer who may notice issues that are not always visible to someone deeply immersed in the complex process of their own research.
That said, I appreciate the modest changes made to the text more than the defensive tone directed at the reviewer. I believe that the current version of the manuscript possesses the necessary quality and clarity to be published and adequately understood by its intended audience.
Author Response
Thank you for your review.